# Recent changes in Pan-Arctic sea ice, lake ice, and snow on/off timing

Alicia A. Dauginis[1], Laura C. Brown[1]

[1]Department of Geography, University of Toronto Mississauga, L5L 1C6, Canada

Correspondence to: LC Brown (lc.brown@utoronto.ca)

**Abstract.** Arctic snow and ice cover are vital indicators of climate variability and change, yet while the Arctic shows overall warming and dramatic changes in snow and ice cover, the response of these high-latitude regions to recent climatic change varies regionally. Although previous studies have examined changing snow and ice separately, examining phenology changes

across multiple components of the cryosphere together is important for understanding how these components, and their response to climate forcing, are interconnected. In this work, we examine recent changes in sea ice, lake ice and snow together at the pan-Arctic scale using the Interactive Multisensor Snow and Ice Mapping System 24 km product from 1997 – 2019, with a more detailed regional examination from 2004 – 2019 using the 4 km product. We show overall that for sea ice, trends towards earlier open water (-7.7 d decade[-1], $p < 0.05$) and later final freeze (10.6 d decade[-1], $p < 0.05$) are evident. Trends

towards earlier first snow-off (-4.9 d decade[-1], $p < 0.05$), combined with trends toward earlier first snow-on (-2.8 d decade[-1] $p < 0.05$), lead to almost no change in the length of the snow-free season, despite shifting earlier in the year. Sea ice-off, lake ice-off and snow-off parameters were significantly correlated, with stronger correlations during the snow/ice-off season compared to the snow/ice-on season. Regionally, the Bering and Chukchi Seas show the most pronounced response to warming, with the strongest trends identified toward earlier ice-off and later ice-on. This is consistent with earlier snow/lake

ice-off and later snow/lake ice-on in west and southwest Alaska. In contrast to this, significant clustering between sea ice, lake ice and snow-on trends in the eastern portion of the North American Arctic show an earlier return of snow and ice. The marked regional variability in snow and ice phenology across the pan-Arctic highlights the complex relationships between snow and ice, and their response to climatic change, and warrants detailed monitoring to understand how different regions of the Arctic are responding to ongoing changes.

## 1 Introduction

The cryosphere is the second largest component of the global climate system after the ocean, exerting significant effects on the Earth's energy balance, atmospheric circulation, and heat transport (Lemke et al. 2007; Callaghan et al. 2011; Derksen et al. 2012). The relevance for climate variability and change is based on physical properties, such as high surface reflectivity (albedo) and latent heat associated with phase changes, both of which have a strong impact on the surface energy balance (Lemke et al. 2007). The extent and duration of snow and ice cover have direct feedbacks to the climate system as they strongly influence planetary albedo (Rahmstorf, 2010; Derksen et al. 2012). Seasonal snow and ice cover are also important for Arctic ecosystems as they rely on snow and ice cover for feeding, transportation, and habitat (Dersken et al. 2012). Additionally, the traditional ways of life of many Northern residents depend on snow and ice cover for sources of food, transportation, and economic activities (Derksen et al. 2012). Recent assessments reveal strong linkages between decreasing snow and ice cover and increasing temperatures in the Arctic (e.g. Hernandez-Henriquez et al. 2015; Johannessen et al. 2016; Druckenmiller and Ritcher-Menge, 2020). Reductions in sea ice extent, decreases in snow cover duration, and earlier melt onset in Arctic and sub-Arctic lakes have been reported (Serreze and Stroeve, 2015; Surdu et al. 2016; Mudryk et al. 2018). Arctic surface air temperatures in 2019 were the second highest in the 120-year (1900 – present) observational record (Druckenmiller and Richter-Menge, 2020), and are projected to continue to increase well into the twenty-first century (Overland, 2020). Though the Arctic as a whole is undergoing climatic change, observations are often marked by regional differences tied in part to global connections via the atmosphere and ocean (Druckenmiller and Richter-Menge, 2020). For example, sea ice in the Alaska/Russia region has shown large reductions in extent over the past decade, which has been linked to strong warming and large sea surface temperature anomalies in this area (Druckenmiller and Richter-Menge, 2020; Perovich et al. 2020). The Canadian Arctic Archipelago (CAA), however, has been shown to exhibit earlier freeze trends during recent years (e.g. Dauginis and Brown, 2020) and weaker trends toward earlier melt onset compared to other Arctic regions (e.g. Mahmud et al. 2016; Marshall et al. 2019). Furthermore, the effect of warming on sea ice dynamics in this region can be counterintuitive as warming could result in increased ice import from the Arctic Ocean into the CAA (Melling, 2002; Howell and Brady, 2019; Moore et al., 2021). Therefore, monitoring Arctic snow and ice cover is critical to improve our understanding of this complex and variable region in the context of climate variability and change.

Monitoring Arctic snow and ice cover largely relies on the use of satellite observations, as ground-based observations are constrained by limited in situ data, large gaps and biases in surface observing networks, and limited geographic coverage (Brown et al. 2010; Brown and Duguay, 2011). Satellite-based microwave data are most used in snow and ice monitoring as they provide information regardless of solar illumination and cloud cover (Brown et al. 2014). Microwave measurements have been used to estimate snow (both on land and on sea ice) and ice melt and freeze onset (e.g. Howell et al. 2006; Yackel et al. 2007; Wang et al. 2011; Zheng et al. 2017; Bliss et al. 2017; Bliss et al. 2019) at various spatial resolutions ranging from 6.25 to 25 km (Brown et al. 2014). The Special Sensor Microwave/Imager (SSM/I) and Scanning Multichannel Microwave Radiometer (SMMR) passive microwave datasets have been widely used in snow and sea ice mapping (e.g. Cho et al. 2017;

Lynch et al. 2017; Crawford et al. 2018). Passive microwave data are well-suited for snow and ice monitoring due to all-weather imaging capabilities and long available records (since the late 1970s), though the coarse resolution (25 km) limits their application and reduces the accuracy of estimates (Derksen et al. 2004; Gao et al. 2010; De Lannoy et al. 2012). There are well-documented uncertainties in using passive microwave measurements to retrieve snow water equivalent and snow cover extent due to differences in snow and surface cover properties (e.g. snow depth, snow grain size, topography, vegetation), which influence microwave emission and backscatter (Brown et al. 2010; Park et al. 2012; Tedesco et al. 2015). The coarse spatial resolution also limits the ability of the sensors to resolve small leads and polynyas and can result in errors near coastal areas due to pixel-based land contamination (Howell et al. 2006; Brown et al. 2014). Johnson and Eicken (2016) note that strong brightness temperature contrasts across pixels can result in falsely high estimates of sea ice concentration, particularly during the summer when there is open water near coastal areas, while in contrast, it is known that that passive microwave data can underrepresent sea ice coverage when liquid water is present (melt ponds on the ice, or wet snow) (e.g., Meier, 2005). SMMR and SSM/I are less commonly used in lake ice applications as the spatial resolution limits analyses to large lakes only. Additionally, the 85 GHz channel is susceptible to considerable atmospheric interference, and the 25 km spatial resolution can result in large differences in water/land brightness temperatures (Cavalieri et al. 1999; Howell et al. 2009).

Optical remote sensing data have also been used to monitor Arctic snow and ice cover (e.g. Nitze et al. 2017; Young et al. 2018) as they provide an improved spatial resolution (e.g. 500 m Moderate Resolution Imaging Spectroradiometer Snow Product) compared to passive microwave data. The use of optical imagery is limited to the spring and summer months in high-latitude regions as there is no source of illumination during late fall and winter due to polar darkness. Additionally, the poor temporal resolution of some optical data (e.g. 16 days for Landsat, 8-day MODIS snow product) can introduce uncertainty and inaccuracy into estimates of snow conditions on the Earth's surface. Active microwave data have been used successfully in snow (e.g. Brown et al. 2007), sea ice (e.g. Mortin et al. 2014), and lake ice (e.g. Howell et al. 2009) applications. Active microwave algorithms using synthetic aperture radar (SAR) provide high resolution (20 to 100 m) retrieval of snow and ice parameters (e.g. Surdu et al. 2016; Zhu et al. 2018; Howell and Brady, 2019). SAR estimates of snow and ice cover provide the highest spatial resolution compared to other products, however the moderate temporal resolution, narrow swath width, and limited image availability across the Arctic limits the application of SAR to smaller geographic regions (Brown et al. 2014; Howell et al. 2019). Multisensor approaches exploiting advantages of microwave and optical sensors have been used to estimate snow thickness on first year sea ice (e.g. Zheng et al. 2017) and to resolve leads and polynyas at an improved spatial resolution (e.g. Ludwig et al. 2019). The all-weather capabilities of microwave data combined with high temporal resolution of optical imagery can improve estimates of snow and ice parameters in the Arctic.

An alternative approach to snow and ice mapping is the use of the National Ice Center Interactive Multisensor Snow and Ice Mapping System (IMS) product. IMS is created using a variety of multi-sourced datasets (e.g. optical imagery, microwave data, ancillary data) and provides daily maps of snow and ice cover at 24 km, 4 km, and 1 km spatial resolutions (Ramsay 1998; Helfrich et al. 2007). The daily temporal resolution and all-weather monitoring capabilities make IMS suitable in snow cover applications (e.g. Brubaker et al. 2005; Chen et al. 2012; Yu et al. 2017) and lake ice monitoring on large lakes

(e.g. Brown and Duguay, 2012; Duguay et al. 2012, 2013, 2014, 2015; Duguay and Brown, 2018). Though not commonly used in sea ice applications, Brown et al. (2014) show that IMS is advantageous over several automated algorithms for monitoring sea ice phenology. IMS is also able to improve sea ice estimates by reducing land contamination and better representing coastal regions compared to passive microwave estimates (Brown et al. 2014), and to resolve finer-scale details between narrow ocean channels (Dauginis and Brown, 2020). This work expands on the work of Dauginis and Brown (2020) and examines changes in sea ice, lake ice, and snow phenology from 1997 – 2019 across the pan-Arctic. The objectives of this paper are to 1) assess changes in sea ice, lake ice, and snow phenology from 1997 – 2019 across the pan-Arctic and 2) analyze regional changes in snow and ice phenology during more recent years (2004 – 2019) across the pan-Arctic.

## 2.0 Methodology

### 2.1 Study regions

In this study, regions north of 56º were considered when examining pan-Arctic snow and ice phenology (Figure 1) to include much of the southern limits of the sea ice in the Bering Strait and large Arctic lakes that can be resolved using IMS. For the second section of the results, a regional approach was taken. For snow and lake ice, phenology parameters were considered on a hemispheric scale (i.e. North America and Eurasia). Further regional subdivisions are provided in Table 3 for the snow and lake ice trends. For sea ice, phenology parameters were examined in three broad regions (with some subregions included in Table 3): Canadian Arctic, Alaska/Far East Russia, and Eurasian Arctic. 'Canadian Arctic' includes Baffin Bay, Hudson Bay, and the CAA; 'Alaska/ Far East Russia' includes the Beaufort, Chukchi, and Bering seas; 'Eurasian Arctic' includes the East Siberian, Laptev, Kara, Barents, and Greenland seas. These regions were grouped based on similar trends in phenology parameters and differences in climate and weather characteristics at the hemispheric scale.

### 2.2 Data

Snow and ice data were obtained from the Interactive Multisensor Snow and Ice Mapping System archived at the National Snow and Ice Data Centre (2004 – present, https://nsidc.org/data/G02156/versions/1) as well as from the National Ice Centre (2014 – present, https://usicecenter.gov/Products/ImsHome). IMS is an operational product used to map daily snow and ice cover over the Northern Hemisphere at 1 km (2014 – present; not used in this study due to the limited time series), 4 km (2004 – present), and 24 km (1997 – present) spatial resolutions. Analysts use a variety of multi-sourced datasets (for a complete list of data sources, see National Snow and Ice Data Center, https://nsidc.org/data/g02156) to subjectively produce maps with discrete values assigned to land, snow-covered land, water, and ice. Snow mapping primarily relies on visible imagery; however, if visible imagery is unavailable due to cloud occlusion or low solar illumination, microwave data is used instead (Helfrich et al. 2007; Brown et al. 2010). As misidentification errors associated with microwave data can occur, analysts rely more on snow climatology compared to microwave data to estimate high latitude snow cover during winter months (Chang et al. 1996; Foster et al. 2005; Helfrich et al. 2007; Derksen 2008; Brown et al. 2010). Ice cover analysis primarily relies on

AVHRR or MODIS observations, however microwave-based retrievals and ice climatology are used when visible imagery is
unavailable, with microwave retrievals representing approximately 30-35% of the ice cover input (Helfrich et al. 2007). IMS
has been shown to outperform data from traditional passive microwave products (AMRS-E, SSM/I, SSMI-SSMIS) for both
the timing and extent of first open water in the Arctic (Brown et al., 2014). For example, through the Barrow Strait in the CAA,
the ability of the 4 km IMS data to resolve narrow channels lead to 17% more open water detected than with SSM/I, and 35%
more open water than detected with AMSR-E, validated with RADARSAT-1 (Brown et al., 2014). Overall, most pixels
compared between IMS and the two passive microwave datasets for first open water dates were within ± 5 days, with a greater
percentage of the pixels in the categories beyond the ± 5 days identifying open water earlier with IMS than the other products
(Brown et al., 2014). IMS has been shown to map higher snow cover fractions during the spring melt period than other snow
products (Brown et al. 2010; Frei and Lee, 2010), but is reported to have mostly between 80-90% agreement with other snow
products during the winter season of non-arctic North America, with better agreement in the later part of the winter season
when deeper and more extensive snow cover is present (Chen et al., 2012). For lake ice, the 4 km IMS product occasionally
identifies earlier lake ice-on dates in regions of prolonged cloud cover (e.g., northern Quebec, Canada), though both ice-on
and -off timing detected using IMS are significantly correlated with, and comparable to, phenology dates extracted from the
MODIS Snow Cover product (Brown and Duguay, 2012).

Temperature data (2 m) are from the European Centre for Medium-Range Weather Forecasts (ECMWF) ERA5 global
reanalysis (CS3, 2017 DOI: 10.24381/cds.f17050d7; Hersbach et al., 2020) and were compared to changes in snow and ice
phenology. ERA5 provides coverage of the entire Arctic at a spatial resolution of approximately 31 km (0.25º). Monthly
temperature data were used to calculate temperature trends from 2004 – 2019. Temperature data near the surface (1000 mb)
were reported to have a 0.89 K difference from radiosonde observations, and the ensemble spread is quite low at ~ 0.4 K or
less, from 1979 - 2018, which can be used as an indicator of uncertainty (Hersbach et al., 2020). Compared to radiosonde
temperature profiles in the Fram Strait, ERA5 had the smallest bias ($\leq 0.3°$) and RMSE ($\leq 1.0°$), and highest correlation
coefficients ($\geq 0.96$) over four other reanalysis datasets tested (ERA-Interim, JRA-55, MERRA-2, CFSv2) (Graham et al.,
2019). The 2 m air temperature in ERA5 has improved fit to observations in the Arctic compared to its predecessor ERA-
Interim (Hersbach et al., 2020), though Wang et al. (2019) show ERA5 has a warm bias over sea ice compared to observation
data from buoys. The identified warm bias is stronger in the cold season, particularly when the 2 m air temperature is below -
25°C (daily mean value of 5.4°C), however monthly mean differences between ERA5 and buoys are ~2°C or less through all
months other than March, April, and May (Wang et al., 2019). Regionally, ERA5 performs best in the Central Arctic, followed
by the Pacific Sector; the Atlantic sector shows good agreement only while the 2 m temperatures are above -25°C (Wang et
al., 2019). We acknowledge that some small potentially spurious regions of opposite trend directions appear in some months
of the temperature trend maps (e.g., February: Eastern Siberia, March, and October: Arctic Ocean) however these data are not
used in a quantitative comparison and therefore do not affect the overall discussion.

Downwelling longwave radiation has been linked to melt onset in the Arctic Ocean (e.g., Mortin et al., 2016). To further explore the linkages in the phenology data, downwelling longwave radiation data from the Extended AVHRR Polar
Pathfinder (APP-x) was obtained from NOAA National Centres for Environmental Information (https://www.ncei.noaa.gov/data/avhrr-polar-pathfinder-extended/access/) (Key et al., 2019). APP-x data is provided as 25 km EASE grid projection, processed for 0400 and 1400 (LST). Due to large areas of missing data between ~ 59 - 64°N, the mean monthly values were created from the 0400 and 1400 separately to avoid averaging errors where data exist for one time and not the other (to avoid skewing the average with the diurnal differences). Some artificial patterns are evident in the data (e.g.,
March, Figure 9c, near the pole), however for the purpose of regional comparisons this is not limiting as this region is not used in quantitative comparisons. Downwelling longwave radiation at the surface is calculated using a neural network to simulate a radiative model (see Key and Schweiger 1998; Key et al., 2016). Downwelling longwave radiation was selected from APP-x rather than ERA5 as the APP-x dataset has been determined as 'climate data record quality' and has been validated against in situ data with a bias of only 2.1 Wm$^{-2}$ and RMSE of 22.4 Wm$^{-2}$ (with the higher RMSE values attributed to differences in
surface snow fall between the sampling site and the 25 km x 25 km area represented) (Key et al, 2016).

### 2.3 Methodology

The 24 km and 4 km IMS products were used to examine changes in snow and ice phenology dates across the pan-Arctic following the methodology of Brown et al. (2014) and Dauginis and Brown (2020). For each pixel, consecutive days of IMS imagery were compared to determine the first and last changes between snow/ice and land/water to determine the timing
of the snow/ice-on and off parameters examined. The phenology parameters used in this study and their definitions can be found in Table 1. Only the first and last change from ice/water and vice versa are tracked for this work, giving first and final dates of change. In sea ice regions dominated by thermodynamics, there is little difference between first and final timing, whereas in more active ice regions there could be a more notable difference between the first and final timings as the ice moves past that pixel. Most lakes are dominated by thermodynamics and return similar first and final dates, however, lakes with more
ice motion (e.g., Lake Onega and Ladoga) may show a difference in their timings. For snow, warmer regions where more frequent snowmelt occurs tend to show a larger variation in first and final dates compared to the northern regions where the snow typically remains on the ground for the season. Open water duration and snow-free duration are defined as the time between the final change in the spring to the first change in the fall (WCI$_S$ to FO$_S$, and last_s$_{OFF}$ to first_s$_{ON}$). The 24 km IMS product was used to examine trends in mean snow and sea ice phenology dates across the pan-Arctic from 1997 – 2019. For
lake ice, only the 4 km IMS product (2004 – 2019) was used since the 24 km product can only detect very large lakes (Figure 2). In addition to detecting more lakes, the 4 km IMS product can also provide more detailed information on lake ice phenology within each lake, as shown in Figure 2.

To investigate the relationship between phenology parameters and temperature, and phenology parameters and downwelling longwave radiation (two important drivers of phenology in the arctic), regional correlations between variables
were examined using Spearman's rank correlation coefficient ($\rho$) as this method describes the overall strength of the

relationship between two variables and does not require data to follow independent normal distributions (non-parametric) (Hauke and Kossowski, 2011). All data was projected to match the IMS data, and the centre point of all grid cells within each specific region were used for the analysis. Datasets were detrended prior to correlation analysis to ensure relationships were not a result of a shared trend, but rather driven by actual relationships between variability in phenology parameters and

temperature (Pizzolato et al. 2014). Data were detrended using the "pracma" package in R (https://CRAN.R-project.org/package=pracma) which removes the linear trend from a given dataset by computing the least-squares fit of a straight line to the data and subtracting the resulting function from the data (Borchers, 2019). The detrended data were then used to calculate Spearman correlation coefficients between phenology parameters and temperature.

To evaluate spatial trends in snow and ice phenology, 4 km IMS phenology dates, 2 m air temperature data, and

downwelling longwave radiation were analyzed using the "zhang" method of trend analysis, available in the "zyp" package in R (Bronaugh and Werner 2019). This method of trend analysis was proposed by Zhang et al. (2000) and has been successfully used to represent trends in temperature and precipitation (Zhang et al. 2000) lake ice phenology (Murfitt and Brown 2017) and sea ice and snow phenology (Dauginis and Brown, 2000). The "zhang" method is suitable for analyzing spatial trends in this study as it employs non-parametric tests and accounts for autocorrelation. The linear trend is removed from the time series if

it is significant, and the autocorrelation computation repeats until the differences in the estimates of the slope and autoregressive model in two consecutive iterations is smaller than 1% (Bronaugh and Werner 2019). The Mann-Kendall test is applied to the resulting time series and the Sen's slope of the trend is computed (Bronaugh and Werner 2019). The final result is the Sen's slope (amount of increase or decrease) at each location over the given time period, as well as the significance of each trend (Bronaugh and Werner 2019). Interannual and regional variability in snow and ice conditions will inherently

affect phenology parameters, particularly for sea ice, which may not entirely clear out of some regions in a particular season leading to no ice-off or -on phenology detected for that year (Dauginis and Brown, 2020). Pixels with less than 14 years of phenology data (e.g. regions where ice-off only occurs occasionally) are treated as No Data, meaning the spatial extent of the trend examination represents the geographic region where snow/ice-off has occurred in at least 14 of the last 16 years.

Finally, clustering in the trend data was explored using local indicators of spatial association (Anselin, 1995) through

ESRI ArcGIS. Clusters of spatially statistically significant trends of high and low trend strengths were mapped. Clusters crossing the shorelines indicate significantly clustered trends between the sea ice and snow or lake ice phenology parameters and show regions of interest where the phenology variables were responding with similar trend strength over the study period.

## 3 Results and Discussion

## 3.1 Trends and Correlations

Mean snow, sea ice, and lake ice phenology dates across the pan-Arctic are shown in Figure 3 (4 km IMS, 2004 – 2019). Mean snow, sea ice, and lake ice phenology trends for the 24 km (1997 – 2019) and 4 km (2004 – 2019) IMS products are shown in Figure 4. Overall, the pan-Arctic shows trends toward a longer snow and ice-free season (Figure 4) from 1997 – 2019, with trends toward earlier snow-off and ice-off and later freeze detected. While the annual variability is similar between the 24 km and 4 km mean phenology dates, a difference of 3.5 days later for ice-off and 3.4 days earlier for ice-on (average) is evident in the sea ice phenology as a result of the resolution differences, mainly attributed to the improved ability of the 4 km product to resolve smaller-scale features and changes in the ice cover extent than the 24 km product can detect (e.g. leads, polynyas, near-shore conditions, and changes at the ice edges) (Brown et al. 2014; Dauginis and Brown, 2020). The overall agreement between the products is < 1 day for the snow phenology dates.

Sea ice open water dates both show significant negative (earlier) trends (Figure 4a), with a larger negative trend detected for first open water (FOW$_S$, -7.72 d decade$^{-1}$, $p < 0.05$) compared to water clear of ice (WCI$_S$, -3.31 d decade$^{-1}$, $p < 0.05$). Snow-off dates show similar trends to ice-off parameters, with both first snow-off (first_s$_{OFF}$, -4.90 d decade$^{-1}$, $p < 0.05$) and final snow-off (final_s$_{OFF}$, -3.21 d decade$^{-1}$, $p > 0.05$) becoming earlier (Figure 4b). Trends for lake ice first open water and water clear of ice dates from 2004 – 2019 are negative (-0.76 and -0.02 d decade$^{-1}$, $p > 0.05$, though neither are statistically significant. We acknowledge that the 16-year time series (Figure 4c) does not provide a comparative time-span to the other trends examined; however, it should be noted that the direction of the trends are negative (earlier) and therefore follow a similar pattern observed in snow and sea ice trends during the 1997 – 2019 melt season.

Sea ice freeze onset (FO$_S$) shows a slightly positive (later) trend (0.36 d decade$^{-1}$, $p > 0.05$), while the continuous ice cover (CIC$_S$) trend is much larger and statistically significant (10.60 d decade$^{-1}$ $p < 0.05$) (Figure 4d). Both first and final snow-on (first_s$_{ON}$, final_s$_{ON}$) trends are negative, indicating that the pan-Arctic is seeing earlier snow onset over the 1997 – 2019 study period. First snow-on is becoming earlier by 2.79 d decade$^{-1}$ ($p < 0.05$) while final snow-on is becoming slightly earlier by 0.64 d decade$^{-1}$ ($p > 0.05$) (Figure 4e). Lake ice freeze onset (FO$_L$) and continuous ice cover (CIC$_L$) exhibit trends toward later freeze (4.97 and 4.44 d decade$^{-1}$, $p > 0.05$; Figure 4f), and although caution should be taken with the short timespan, it should again be noted that lake ice freeze dates show an overall shift toward later freeze.

Overall, snow and ice cover are coming off earlier across the pan-Arctic, while trends during the freeze season vary for sea ice, lake ice, and snow. Earlier sea ice water clear of ice dates contribute to longer open water duration detected across the pan-Arctic (4.85 d decade$^{-1}$, $p > 0.05$; Figure 5). Non-significant trends are detected in lake ice parameters, with the resulting open water duration in Arctic lakes increasing by 6.86 d decade$^{-1}$ from 2004 – 2019 ($p > 0.05$) (Figure 5). Almost no trend in snow-free duration is identified (-0.27 d decade$^{-1}$, $p > 0.05$; Figure 5), despite first snow-off trending significantly earlier (Figure 4b).

Examining the pan-Arctic links between the phenology parameters shows that while the first open water and first snow-off dates are not significantly correlated, the final snow/ice-off parameters are (sea ice water clear of ice and final snow-

off dates, $\rho = 0.46$ and 0.64, $p < 0.05$, for 24 and 4 km IMS products respectively) (Table 2). During the snow/ice-off season, lake ice first open water ($FOW_L$) and water clear of ice ($WCI_L$) dates are significantly correlated with their equivalent snow and sea ice-off parameters from 2004 – 2019 (Table 2). Stronger relationships are identified between lake ice and sea ice off parameters ($\rho_{FOW\ Sea\ Ice\ and\ Lake\ Ice} = 0.62$ and $\rho_{WCI\ Sea\ Ice\ and\ Lake\ Ice} = 0.72$, $p < 0.05$) compared to lake ice and snow ($\rho_{first\ snow-off\ and\ FOW\ Lake\ Ice} = 0.55$ and $\rho_{final\ snow-off\ and\ WCI\ Lake\ Ice} = 0.51$, $p < 0.05$). Snow-on dates show small positive correlations with sea ice freeze parameters, though none are statistically significant (Table 3). No significant correlations are detected between lake ice/sea ice and lake ice/snow parameters during the freeze season, though similar to the snow/ice-off season, stronger correlations are detected between lake ice and sea ice freeze compared to lake ice-on and snow-on (Table 2).

Examining snow and ice cover at the pan-Arctic scale provides important information on how the cryosphere is responding to climate change as a whole, however the large degree of spatial variability warrants further investigation into snow and ice conditions at regional scales. For example, Dauginis and Brown (2020) demonstrate that the CAA is responding differently to warming compared to other regions of the Arctic; their findings show later summer clearing of ice and earlier sea ice freeze and snow onset since 2004 (due, at least in part, to increased ice dynamics through the CAA), in line with findings from previous studies that showed no significant trends toward earlier sea ice melt onset dates in the CAA (e.g. Mahmud et al. 2016; Marshall et al. 2019). Other Arctic regions have shown significantly earlier sea ice melt onset, Barents Sea 8.2 decade[-1], Kara Seas 5.1 decade[-1], Baffin Bay 6.6 decade[-1] and Greenland Sea 7.1 decade[-1] (Stroeve and Notz, 2018). The response of snow cover to changes in climatic and hydrologic regimes also varies regionally, with northern Canada and eastern Siberia experiencing increased snowfall, while Scandinavia and regions around the Greenland ice sheet are experiencing increasing rainfall (Box et al. 2019). Additionally, ice cover duration in Arctic lakes since 2004 shows interannual and regional variability, with lakes in western Russia showing anomalies ranging from 59 days shorter to 57 days longer, while smaller anomalies were identified in Canadian Lakes (Duguay and Brown, 2018). Therefore, the following section will examine regional variability in sea ice, lake ice, and snow phenology from 2004 – 2019 using the 4 km IMS product as the higher spatial resolution (compared to the 24 km product) allows finer-scale changes in snow and ice cover to be detected.

## 3.2 Regional Variability

### 3.2.1 Snow and Ice-off season

Short-term trends in sea ice, snow, and lake ice phenology from 2004–2019 are presented in Figures 6 (snow/ice-off) and 7 (snow/ice-on) along with maps identifying significant local clustering in the trends, indicating similar trends between the phenology parameters. Median values of the spatial trends in Figures 6 and 7 for regions defined in Figure 1 are reported throughout the following section and included in Table 3. Correlations with 2 m air temperature for the three main sea ice regions and two main snow/lake ice regions (Figure 1) are presented in Table 4. Overall, sea ice, snow, and lake ice show tendencies toward earlier melt, with the exception of 1) Eurasian snow-off parameters, which show little change from 2004–2019 compared to other Arctic regions and 2) sea ice first open water in the Canadian Arctic. The Alaska/ Far East Russia

region exhibited the largest trends toward earlier sea ice-off (median$_{\text{First Open Water Sea Ice}}$ = 23 days) and North America showed larger trends toward earlier snow-off and lake ice-off compared to Eurasia (North America: median$_{\text{First snow-off}}$ = 8 days, median$_{\text{First Open Water Lake Ice}}$ = 4 days; Eurasia median$_{\text{First snow-off}}$ = 0 days, median$_{\text{First Open Water Lake Ice}}$ = 1 day).

Canadian Arctic

In the Canadian Arctic, sea ice has a wide range of ice-off timing, spanning March in the far southern reaches, to early May in the north for near shore, polynya and lead regions, to late August clearing from the channels of the CAA – in the portions where clearing occurs. The majority of the region experiences ice-off conditions through June, July and August, and these months show significant (negative) correlations between regional mean sea ice-off dates and the regional mean 2 m temperature (as well as May and WCI$_S$, Table 4), indicating that earlier sea ice-off dates here are strongly related to air temperature during the ice-off season. Downwelling longwave radiation shows significant (negative) corelations to WCI$_S$ in September, which is also when ice can clear from the channels of the CAA (where it clears) (Table 5). For the North American Arctic region, snow-free timing ranges from mid-April in the south to mid-July in the north. Regional mean first snow-off dates are significantly correlated with both 2 m temperature and downwelling longwave radiation values for April and May, while final snow-off is significantly correlated only with May (when much of the mainland area of Canada becomes snow-free, Figure 3). Lake ice-off timing spans from April in the south to July in the northern islands (very few lake pixels experience ice-off in August, other than Lake Hazen). Both first and final lake ice-off regional mean dates are significantly correlated with May and June 2 m air temperatures, while the bulk of the ice-off dates in this region are through June and July; it is established that there is lag in air temperatures crossing the 0 °C isotherm and the timing of lake ice-off; up to about a month on average for lakes across Canada (Duguay et al., 2006). Regional mean downwelling longwave radiation through this region shows significant (negative) correlations between first and final open water in both June and July (as well as May WCI$_L$, Table 5). A strong example of the lag between lake ice-off and snow-off can be seen using Great Bear and Great Slave Lakes compared to their surrounding areas (Figure 3): snow-free timing occurs here in May, while the ice remains on the large lakes until June/early-July due to the extra energy required to melt ice vs. snow.

Sea ice is clearing out of the Canadian Arctic earlier, while the first detection of open water (FOW$_S$) shows a later trend, albeit with considerable regional variability. In the CAA, earlier first open water is detected (median = 4 days), though changes toward earlier water clear of ice are mostly confined to the southern channels, where temperature increases are larger in August and September (Figure 8h, i). The significant clustering between water and land pixels in the northern CAA (Figure 6c, d), with predominantly later trends for both open water and snow-off (no lakes are large enough in this area to be detected by IMS), indicates ice and snow are responding similarly in this region. For example, the Eastern Parry Channel and the surrounding area (Cornwallis Island, Bathurst Island, and Northern Somerset Island) shows significant clustering between the sea ice and snow trends (Figure 6c, d), further highlighting this region of later ice/snow-off (Dauginis and Brown, 2020). The Baffin Bay / Davis Strait region overall shows a median trend of 1 day earlier, however, the northern portion (Baffin Bay) and southern portion (Davis Strait) show opposite trend directions (9 days earlier vs. 24 days later, respectively) (Figure 6a, Table

3). Warming trends are identified over the northern region of Baffin Bay in July and August ranging from 0.01 to 3ºC (Figure
8g, h) where the notable trends toward earlier ice-off are detected (Figure 6a, b). Later first open water trends are evident for
Hudson Bay (median = 2 days), with earlier water clear of ice trends (median = 7 days). The median temperature increase
during July over Hudson Bay is 0.55ºC, though while the majority of the northern and western portions show warming trends,
the eastern and more southern portions exhibit cooling (Figure 8g). These regions of cooling correspond with the region of
predominantly later $FOW_S$ trends (median = 2 days). Significant clustering is shown in the ice and snow-off trends along the
southern stretches of Hudson Bay and near-onshore regions (Figure 6c, d), indicating links in the response of ice and snow in
those regions.

Earlier trends for snow and lake ice-off parameters are detected across North America. Looking at the western
mainland areas of the Canadian Arctic, snow-off trends are predominantly earlier over the 2004–2019 period (median first_$s_{OFF}$
= 11 days, median final_$s_{OFF}$ = 10 days, Table 3). Links between the sea ice trends and the snow/lake ice trends are evident in
the region spanning east from Victoria Island into the central mainland Arctic region, where significant local clustering is
identified, with more clustering evident during first snow and ice-off events. Lake ice first open water and water clear of ice
are both trending earlier here as well (median = 5 days for both), with larger trends detected in the eastern portions, likely
related to strong warming over the region in May and June (Figure 8e, f). Examining Great Slave Lake and Great Bear Lake
indicates that trends for water clear of ice are 4 and 8 days (median values) earlier, consistent with negative ice cover duration
anomalies (shorter ice cover duration) for 9 of the last 14 years identified by Duguay and Brown (2018). While the snow trends
surrounding these large lakes are also earlier, they are not significantly clustered with the lake ice trends, with the exception
of the eastern portion of Great Bear Lake, and a small portion of the western edge. The western portion of Great Slave Lake
shows mostly earlier ice-off trends, while the surrounding snow trends show very slight tendency towards later snow-off,
which aligns with both cooling temperature trends and less downwelling longwave radiation in that region (e.g. June, Figure
8f, 9f). Interestingly, the temperature and radiation trends here clearly show the effect of the lakes on their surroundings with
contrasting trends for the lakes compared to land to the west. Lake ice-off dates in northern Quebec (bordered by Hudson Bay
and Baffin Bay) show later trends, with both ice-off parameters showing median trends of 9 days later, which corresponds to
a widespread cooling pattern over northern Quebec in July from 2004–2019 (Figure 8g). North of this region, the two large
lakes on Baffin Island, Nettilling Lake and Amadjuak Lake, show trends for first open water are 3 days (median values) earlier
for both lakes, though from 2004–2018 these lakes have shown positive ice cover duration anomalies for 7 of the last 14 ice
seasons, with most of the longer ice cover duration anomalies observed during the last six seasons (Duguay and Brown, 2018).
Lake Hazen, in the far north, indicates trends towards earlier first open water and water clear of ice overall; however, while
the eastern portion of the lake shows earlier ice-off trends, the western portion does indicate later ice-off trends (with ice cover
remaining latest on the western portion of the lake for several years).

Alaska / Far East Russia

Mean sea ice-off timing in this region is quite different for the Bering Sea (mainly April through May, with some late March first open water) and the Chukchi / Beaufort Seas (mainly late August through mid-September). Significant (negative) correlations between sea ice-off dates and air temperature are identified in this broad region only for $FOW_S$ in May, however warming patterns are present over the Bering/Chukchi Seas for almost all months since 2004 (Figure 8). Interestingly, both ice-off parameters show positive correlations with September temperatures – likely an artifact of the large region compared, as only the far northern sections of this region experience ice-off in September (Table 4). Downwelling longwave radiation in this region is significantly correlated with the ice-off parameters in both March and May and is likely a reflection of the conditions initiating melt onset (e.g. Mortin et al., 2016), as timing of onset is correlated with timing of ice retreat (Stroeve et al., 2016). Snow-free timing in this region is predominantly through late April, early May in the Interior, and mid-June on the North Slope and Chukchi Peninsula, with regions of higher elevations not becoming snow-free until August. Lake ice-off timing spans from May in the southern coastal regions, to July in the northern coastal regions. This land region is included in the North American Arctic snow and lake ice correlations, discussed in the previous section.

Sea ice in the Alaska/ Far East Russian coastal region shows large trends toward earlier ice-off, with first open water trends indicating 30 days earlier in the Beaufort Sea, 25 days earlier in Chukchi Sea, and 34 days earlier in the Bering Sea (median values, with slightly larger values for $WCI_S$). The Chukchi and Bering Seas have shown larger sea surface temperature warming trends in August compared to the Arctic-wide August mean, and September sea ice extent in the Chukchi Sea was well below the 1981 – 2010 median in 2012, 2018, and 2019 (Druckenmiller and Richter-Menge, 2020; Perovich et al. 2020).. First snow-off trends across Alaska/Far East Russia (median = 3 days earlier) are smaller than compared to Canada, though western Alaska shows strong trends toward earlier snow-off. Strong warming over western Alaska from 2004–2019 during April (Figure 8d) may contribute to earlier snowmelt in the region. The median lake ice first open water date shows trends of 18 days earlier in the Alaska/Far East Russia region, though southwestern Alaska shows some of the largest trends toward earlier ice-off (both $FOW_L$ and $WCI_L$) across the pan-Arctic (Figure 6a, Table 3). Local clustering in this western region shows that the on-land and sea ice parameters have statistically significant clustered trends and are changing similarly. Focusing on the more northern regions of Alaska, first snow-off trends across North Slope Alaska (NSA) are towards earlier dates (median = 8 days), though final snow-off trends are considerably smaller (median = 1 day). Lake ice-off across NSA is trending later, with both first open water and water clear of ice showing median dates of 3 days later. Arp et al. (2013) found that the Arctic Coastal Plain (northern Alaska) and Beringia (western Alaska) areas experienced the latest ice-out timing from 2007–2012 compared to other lakes across Alaska, as climatology in these regions is influenced by sea-ice conditions along the Arctic Ocean coast. Though long-term trends (1950 – 2011) indicate earlier ice break-up and shorter ice seasons in NSA (Surdu et al. 2014), the trends toward later ice-off in northern Alaska identified in this study from 2004-2019 (Figure 6a) may be reflecting interannual variability and the complex responses of lake ice to changes in temperature, sea ice, and snow cover conditions. Little to no significant clustering between the snow and ice trends are identified in this region, other than a limited swath of near-shore region, showing the sea ice trends here are stronger than the onshore snow/ice trends.

Eurasia

The broad Eurasian region has sea ice-off timing from April (with a few small regions showing March) in the Barents Sea to September in the Eastern Siberian Sea and Arctic Ocean areas. Both sea ice parameters are correlated with April 2 m air temperature (Table 4) and strong warming patterns can be seen here (Figure 8). $FOW_S$ is correlated with July as well, and ice-off in the Laptev Sea region aligns with positive air temperature trends through July in this region (Figure 3, Figure 8, Table 4). Snow-free timing spans mid-March in the western areas through June along the northern coasts, with some very small regions extending into mid-August on the northern islands. Final snow-off is significantly correlated to 2 m air temperature for April, May and July and final snow-off with downwelling longwave for April and July. Lake ice clearing spans from April in the western European lakes to July in the northern regions (similar to the Canadian Arctic, very few pixels show August ice-free timing in the northern islands), with the 2 m air temperature in April correlated to both ice-off parameters, coinciding with when the lake-rich region in western Europe becomes ice free.

Trends in sea ice first open water range from 11 to 16 days earlier while water clear of ice ranges from 24 to 34 days earlier across the Eurasian Arctic seas (Table 3). Warming patterns over the Eurasian seas are detected in July and August (with the strongest warming over the Laptev and Barents Seas in both months). The earlier ice-off trends detected in this study across the Eurasian Arctic are consistent with Bliss and Anderson (2018), who report negative (earlier) trends in sea ice melt onset across Eurasia from 1979–2017 of -9.45 d decade[-1] (East Siberian), -7.3 d decade[-1] (Laptev), -8.19 d decade[-1] (Kara), -8.47 d decade[-1] (Barents), and -2.37 d decade[-1] (Greenland). Furthermore, earlier ice-off trends in these regions are consistent with large reductions in September sea ice extent in the East Siberian and Laptev Seas from 1979–2016 (Onarheim et al. 2018). Significant local clustering is identified in the sea ice trends near the Laptev and Kara Seas region and the onshore snow/lake ice trends in the northern region of Central Eurasia (Figure 6c, d). Overall, snow-off across the broad Eurasian region shows no trend.. Only NW Eurasia shows notable trends in the snow-off timing, with first snow-off and final snow-off 5 and 3 days later respectively (Figure 6a, b). Crawford et al., (2018) identified links between the reduction of sea ice in the Laptev Sea and the earlier retreat of snow in the Western Siberian Plain, while our results show linked sea ice and onshore trends near the Laptev Sea with earlier snow retreat, but a mix of trend direction in the vicinity of the Western Siberian Plain for snow-off timing. A notable distinction on the Eurasian side of the Arctic is the prominent trend towards later ice-off through the Seas, but the presence of mixed trend directions for the snow-off, and a prominence of clustered later trends through the continental areas. Trends towards decreased downwelling longwave radiation can be seen through this region (Figure 9) as well as cooling trends through September, October, November and January (Figure 8). Lake ice shows variability across Eurasia with ±2 days (median) or less detected, with the exception of Central Eurasia where the ice cover shows trends towards 7 ($FOW_L$) and 9 days earlier ($WCI_L$). Lake Onega (northwest Russia) shows earlier first open water (median = 5 days) and both Onega and nearby Lake Ladoga show earlier water clear of ice trends (Onega median = 6 days, Ladoga median = 9 days). The eastern portion of Lake Ladoga shows very slight trends towards later $WCI_L$, which is in sync with later snow off trends in that region, while the south/western portions show stronger earlier $WCI_L$ off trends, likely related to delayed freeze (resulting in thinner, more easily melted ice) and predominant location of the ice over the season (as the ice forms in the southern shallower potions

of the lake first) (Karetnikov et al., 2017). FOW$_L$ trends were not calculated for Lake Ladoga as the intermittent and moving ice cover through the season presents limitation to the current search algorithm. There is only an 80% chance that Ladoga will experience a full ice cover during the ice season (Karetnikov et al., 2017). From 1955 – 2015, total ice cover duration in Lake Onega decreased by 50 days, though decreases were mostly attributed to delayed freeze (Filatov et al. 2019). Earlier break-up dates have been detected in 40 lakes across Finland from 1963 – 2014 (Kuusisto, 2015), however our recent short-term trends

show that lake ice-off is becoming slightly later (median = 2 days) in Finnish lakes nearby to Lake Ladoga and Onega. Mean 4 km IMS imagery shows that the average break-up dates range from mid-April to mid-May in this region, though temperature trends are only negative (cooler) in southwestern Finland during April and positive (warmer) over all of Finland during May (Figure 8d, e).

### 3.2.2 Snow and Ice-on season

Sea ice freeze onset in the Canadian Arctic shows trends towards earlier timing (median = 11 days), while sea ice within the Alaska/ Far East Russia region and Eurasian regions shows delays in freeze (trends of 8 and 7 days later respectively for freeze onset) (Figure 7a, b). On land, the North American Arctic and the Eurasian regions both show trends towards earlier first snow-on (median$_{North\ America}$ = 8 days and median$_{Eurasia}$ = 9 days) and final snow-on (median$_{North\ America}$ = 3 days and median$_{Eurasia}$ = 7 days), though spatial variability is evident. Unlike the snow/ice-off season where lake ice-off trends were larger over North

America compared to Eurasia, overall trends toward later lake ice freeze onset are larger across Eurasia (median = 8 days) than North America (median = 2 days) (Figure 6c). Local clustering is again evident across the Arctic (Figure 7c, d), however less sea ice/snow clusters are evident in the freeze maps, with the exception of southern Alaska and northern Quebec (Nunavik), particularly for final freeze. A more detailed regional break down follows.

Canadian Arctic

Sea ice freeze in the northern portions of the archipelago takes place through in late September and October.  Most of Hudson Bay freezes back mainly between late November and December; Baffin Bay through November; and the Davis Strait much later with freeze onset generally through February and final freeze through March. Sea ice freeze onset through this broad region shows significant correlations to September and October downwelling longwave radiation, as well as March

2 m air temperatures (freeze is still occurring in the Davis Strait in March), while complete freeze shows correlations to September and November 2 m air temperatures and December downwelling longwave radiation. Increased cloud cover is reported in some sections of the Canadian region through September-November (Boisvert and Stroeve, 2015), which is likely related to the downwelling longwave correlations identified here. Feedbacks between longer open water, increased atmospheric moisture and cloud formation may be related to the delayed freeze – while also resulting in the increased downwelling

longwave radiation here. September, October, and December show some trends in increasing downwelling longwave across the region (Figure 9), while November shows clear decreases through most of the region.  This furthers the suggestion that feedbacks are driving the corelations with downwelling longwave as no correlations are identified in November. Snow returns

to the High Arctic in late August through September, and most of the mainland Canadian areas through October – though final snow on does not occur until November in the southern reaches of the western region. First snow on is significantly correlated

with August, September, and October 2 m air temperatures, and downwelling longwave radiation with September. Most lakes in the Canadian Arctic region freeze in November (with final freeze on the large Great Slave Lake extending into early December in some places), while some lakes in the High Arctic freeze earlier through September and October.  Lake ice freeze is significantly correlated with the November 2 m air temperatures, which coincides with when the majority of lakes freeze.

Earlier sea ice freeze and snow-on trends are detected across the Canadian Arctic. Sea ice shows freeze onset trending

earlier by 8 days in the CAA, and 10 days in Hudson Bay and Baffin Bay (median values) (Figure 6c). First snow-on across the western mainland Canada shows earlier trends (median= 5 days), though delayed snow onset can be identified along north and northwest regions of Canada (south of the Western Arctic Waterway) (Figure 7a, b). Snow-on also shows earlier trends in northern Quebec (median$_{first\ snow-on}$ = 8 days, median$_{final\ snow-on}$ = 16 days) and corresponds to both earlier lake ice and sea ice freeze onset trends identified in this region. A large cooling pattern can be seen over eastern Canada in October which may

contribute to the earlier snow and ice-on dates in this region (Figure 8j). Earlier snow-on dates in the Canadian Arctic are consistent with observed increases in precipitation across all seasons in Canada from 1948 – 2012 (Vincent et al. 2015). Global climate models project increases in Arctic precipitation over the twenty-first century due to enhanced local surface evaporation resulting from sea ice loss; however, recent projections show a shift toward a rain-dominated Arctic, particularly during summer months (Bintanja and Selten, 2014; Bintanja and Andry, 2017). Nettilling Lake shows trends towards earlier freeze

onset (median = 3 days) and continuous ice cover (median = 1 day), though there is considerable variability in freeze-up as the east shows trends toward earlier freeze and west shows trends toward later freeze. Great Slave Lake, Great Bear Lake, Amadjuak Lake and Lake Hazen all show trends toward later freeze onset, consistent with later ice formation across Arctic Lakes from 2002 – 2015 (Du et al. 2017; Derksen et al. 2019). Increases in mean monthly lake surface temperatures in August have been reported to delay freeze-up by 0.3 d decade$^{-1}$ on Lake Hazen from 2000 – 2012 (Lehnherr et al. 2018), and warming

air temperature trends are evident in the August ERA-5 data from 2004 – 2019 as well (Figure 8h). Interestingly, from 2004–2019, freeze onset is showing later trends across most of Lake Hazen, while final freeze is showing earlier trends across the entire lake despite the delayed start to the freeze season. September 2 m temperature trends show slight cooling trends for the pixels covering the lake ranging from -0.6°C to -1.4°C and would correspond to the time of complete freeze over.

Alaska / Far East Russia

Sea ice freeze starts in late September in the northern areas of the Chukchi Sea and ends in late March/Early April in the southern reaches of the Bering Sea. No correlations between temperature and sea ice phenologies in this region were found, however, final freeze for sea ice was significantly correlated with downwelling longwave radiation in January, February and March, coinciding with when most of the freeze is occurring in the Bering Sea. Stronger trends towards increasing 2 m air

temperatures and downwelling longwave radiation are seen in this area through the winter (Figures 8 and 9). This region shows a mix of trends in cloud cover for December-February between 2003-2013 with some decreasing trends over the Bering Sea

and increasing trends over the Chukchi Sea (Boisvert and Stroeve, 2015), and the correlations between ice-on and downwelling longwave identified here are again likely a result of feedbacks taking place. Snow returns to most of the region through September and October, with final snow on occurring as late as November/December in the southwestern coastal regions and as early as late August in the highest elevations. Snow-on and lake ice-on correlations for this region are included in the main North American Arctic region, with first snow-on correlating to August, September, and October 2 m air temperatures and to downwelling longwave radiation in September. November 2 m air temperature is significantly correlated with lake ice freeze, coinciding with when the majority of the lakes in this region freeze, with later freeze through December and January in the southwest coastal regions of Alaska, and September freeze through the NSA.

Delayed sea ice freeze is identified throughout the Alaska/ Far East Russia region, while there is considerable variability in snow onset and lake ice freeze (Figure 7a, b). Trends towards later freeze ($FO_S$ = 6 days median, $CIC_S$ = 8 days) in the Beaufort Sea are consistent with multi-year ice losses and lengthening of the open water season in this region (Galley et al. 2016). The Chukchi and Bering Seas both show trends toward later freeze onset ($median_{Chukchi\ Sea}$ = 8 days and $median_{Bering\ Sea}$ = 27 days) and continuous ice cover ($median_{Chukchi\ Sea}$ = 19 days and $median_{Bering\ Sea}$ = 52 days), with the Bering Sea representing the region with the largest delay in sea ice freeze across the pan-Arctic. During the ice cover season in 2017/2018 the Bering Sea ice extent was lower than any previous winter in the reconstructed or observed record, attributed to warmer sea surface temperatures, delayed freeze, and frequent storms (Thoman et al. 2020). In 2019 the Bering Sea also had extremely low ice cover during the winter and may have acted as a precursor to low summer ice conditions in the Chukchi Sea (Perovich et al. 2020). Sea ice did not completely freeze over in the Chukchi Sea until December 24 in 2019 (approximately a month later than average), with only 2007 and 2016 showing similar freeze patterns since satellite observations began in 1979 (Perovich et al. 2020). Strong warming trends observed over the Bering and Chukchi Seas from October through January (+1 – +6 ℃) likely contribute to the delayed freeze detected (Figure 8 j, k), though no significant correlations between $FO_S$ or $CIC_S$ and 2 m temperatures from October to January are identified in this region. First snow-on trends for Alaska are becoming earlier (median = 5 days), while final snow-on is showing smaller changes (median = 1 day later). The largest trends toward later snow cover are evident in western Alaska, with 3 days later for first snow-on and 22 days later for final snow-on. Wendler et al. (2017) report a 17% increase in mean snowfall across Alaska from 1946 – 2014, with the largest increases occurring in west and southwest Alaska. More snowfall here may be tied to warming in this region identified during almost all months from 2004 – 2019 (Figure 8), as warmer air is able to sustain more moisture which can thus facilitate increases in precipitation (Thackeray et al. 2019). First snow-on in the NSA region is becoming earlier (median = 21 days), whereas final snow-on shows no change overall due to the mix of earlier and later trends throughout the region. Lake ice within the NSA region shows trends toward earlier freeze ($FO_L$ = 8 days and $CIC_L$ = 9 days).

Eurasia

Ice returns to the East Siberian, Laptev and Kara Seas from late September through October, while freeze timing extends through the winter into March for some regions of the Barents and Greenland Seas. The freeze onset timing is significantly correlated with the December 2 m air temperatures, while downwelling longwave radiation is significantly correlated with October and February through this broad Eurasian region. Snow first returns to the islands of the Eurasian Arctic in August, with the continental mainland receiving its snow cover predominantly through September and early October

(with final snow on timing mainly through October). The western regions tend to receive snow later from October into early November, though final snow-on does not typically occur until January in this region. Lakes through eastern and central Eurasia freeze back through October and early November, while the western regions remain mostly ice free until December or January, with freeze on the large Lake Ladoga typically extending into February (though intermittent or partial ice cover is common here). Mean snow-on and lake ice-on do not show correlations to temperature or downwelling longwave radiation in

this broad region, with the exception of final lake ice freeze in February – though this should be interpreted with caution as only Lake Ladoga experiences final freeze that late in the season and the intermittent and partial ice cover on this lake may affect the mean values.

In Eurasia, sea ice freeze onset is becoming later, though the trends in freeze (Figure 7a, b) are smaller in magnitude than the trends for sea ice-off (Figure 6a, b). The later freeze coincides broadly with the regions of earlier ice-off (Figure 6a,

b) where ocean-atmosphere feedbacks will enhance warming through the summer and fall, ultimately delaying freeze (e.g. Stroeve et al., 2014). Warming can be seen over the region from July to November (Figure 8) as well and increasing downwelling longwave radiation in October, and in the western portions December as well (Figure 9). The strongest trends towards later freeze in this region were identified in the Barents, Kara and Laptev Seas, which is in agreement with trends identified from 2000-2013 showing the strongest freeze onset trends in this region in the Barents/Kara Seas followed by the

Laptev Sea (Boisvert and Stroeve, 2015). (Table 4), Snow-on shows predominantly earlier trends across all regions of Eurasia ranging from 4 to 13 days (medians). Increased fall and early winter snow cover through the eastern portions of Eurasia have been linked to decreasing September sea ice cover in the Pacific sector (Ghatak et al., 2010; 2012), and trends towards increasing snow cover extent in October have been linked to increased moisture transport possible because of the delayed sea ice freeze (Yeo et al., 2017). Lake ice shows similar patterns to snow onset, with earlier freeze detected over the Eurasian

regions, with the exception of the Scandinavian/Northern Europe region where freeze onset shows delays of 28 days (median value) and continuous ice cover shows delays of 19 days (median value) later. Large freeze-up anomalies in this region were also identified through previous lake ice research (e.g. Duguay and Brown, 2018), for example the 2017/2018 freeze season showed delayed freeze up by approximately 2 – 5 weeks compared to the 2004 – 2018 mean. Using data from 1890 – 2015, Karetnikov et al. (2017) show that the number of winters with complete freeze over of Lake Ladoga decreased after 1950 and

that the ice season has become shorter. The contrasting trends for delays in freeze onset compared to the earlier snow onset surrounding Lake Ladoga to the north and east also highlight the differences affecting snow-on and ice-on in the freeze season. While the air temperature may be cold enough to sustain a snowfall (first snow-on), the lake water remains warmer through the fall, releasing the stored energy that was absorbed through the extended open water season, as trends here are towards

earlier WCI$_L$ through much of the lake. In fact, for final freeze, although no trends were calculated for Lake Ladoga due to the intermittent ice, the region surrounding the lake shows a delay in final snow-on, and the 2 m air temperature for January (Figure 8a) shows a region of localized warming in the vicinity east of Lake Ladoga and Onega. Whether these localized trends are a direct result of temperature moderation from the lakes, or potentially from feedback processes in the ERA5 data is unknown, but as this is in a common downwind direction of the lakes (Kondratyev and Filatov, 1999), it would suggest lake induced temperature moderation as these lakes tend to freeze through January. Ice cover trends for first open water and complete ice cover are not included for Lake Ladoga in this study as a complete ice cover did not form in several of the examined years. Trends for water clear of ice (full open water) and freeze onset (first detection of ice) are detectable and included in Table 3.

**4 Conclusion**

This paper examined sea ice, snow, and lake ice phenology across the pan-Arctic using the Interactive Multisensor Snow and Ice Mapping System (IMS) snow and ice products. Using IMS, we were able to examine both long-term snow and ice-on/off trends (1997 – 2019) at a 24 km spatial resolution, as well as more recent short-term trends in snow and ice phenology (2004 – 2019) at an improved resolution of 4 km. Our results show that the Arctic is moving toward a longer snow and ice-free season, with trends toward earlier snow/ice-off and later freeze detected. Sea ice showed the largest trends toward earlier ice-off and later freeze, with FOW$_S$ timing becoming earlier by 7.72 d decade$^{-1}$ and CIC$_S$ becoming later by 10.60 d decade$^{-1}$. Lake ice and snow-off parameters are also showing earlier trends, though not as large as those detected for sea ice. Lake ice-off showed significant correlations with snow-off and sea ice-off, while no significant correlations were found between any snow/lake ice/sea ice parameters during the freeze season. This likely reflects the strong influence of surface air temperature on snow and ice off timing, whereas during the freeze season, precipitation patterns play an important role in determining the timing of snow onset and lake size/volume is an important determinant for freeze timing.

Sea ice in the Canadian Arctic is clearing earlier overall, though regional variability does indicate some regions of later clearing, while during the freeze season, sea ice-on trends are predominantly earlier (11 and 9 days median for first and final ice cover), showing opposite trends compared to other regions across the pan-Arctic. Snow-off and lake ice-off show predominantly earlier trends across North America with some regional exceptions in the east. Snow onset also shows earlier trends across North America, with snow-on trends moving earlier by 4 to 16 days. Lake ice shows a mixed response, with later freeze in the west and earlier freeze in the east – reflective of the cooling air temperature trends over the eastern regions. The largest trends toward earlier sea ice-off were detected in the Alaska/Far East Russia region, with trends towards ice clearing a month later. Snow and lake ice-off timing in this region also shows earlier trends, with some of the largest snow and lake ice trends identified in the Western Alaska region. Delays in sea ice freeze were also observed here (trends of 8 and 14 days later for first and final freeze) with much stronger trends in the Bering Sea region, along with delayed snow onset over land and delayed freeze onset in lakes across most of Alaska. Sea ice on the Eurasian side of the Arctic is showing larger trends towards earlier ice-off than later ice-on (roughly a month later ice-off, 1-2 weeks later freeze). No trend in first snow-off dates was

detected in Eurasia and only a small change toward earlier final snow-off (median = 1 day) was identified; though larger trends toward earlier snow-off were detected in northwest Eurasia compared to the east. Lake ice shows a similar east-west pattern, with Lake Ladoga and Lake Onega showing larger earlier first open water trends compared to northeast. In Eurasia the snow cover trends are stronger in the freeze season, and predominantly towards earlier snow-on, while later lake freeze is occurring on the large lakes. Earlier snowfall occurs through this region and is related to feedbacks from the longer open ocean water, however the contrast in trends between snow and lake ice here show that the heat retained in the mixed layer of the lakes through the longer open water season is enough to delay freeze, despite snow falling earlier in the fall/winter.

Overall, stronger trends towards longer open water duration on both the northern oceans and lakes are shown compared to the lack of overall trend in snow-free duration (the earlier snow-off trends are offset by the earlier snow-on trends) (Figure 5). This is in line with stronger Arctic Amplification processes over the Arctic Ocean compared to land (e.g., Miller et al., 2010), with the lower albedo of water allowing for more energy absorption and increased heating than occurs on land. This would apply to lakes as well and is particularly evident in lakes through Alaska with stronger trends towards earlier ice-off and later ice-on compared to snow, as well as in Scandinavia/Northern Europe, where strong opposite trends are shown between later lake ice-on and earlier snow-on. Furthermore, feedbacks related to ocean-atmosphere interactions during the longer open water season are contributing to earlier snow-on timing in some regions. By examining multiple components of the cryosphere together, we can better understand how warming affects snow and ice cover and how these components are interrelated. As the Arctic continues to experience unprecedented change as a response to increasing temperatures, continuous monitoring of changes in snow and ice cover is essential to improve our understanding of climate variability and changes occurring at not only the pan-Arctic scale but at the regional-scale as well.

**Acknowledgements**

Funding for this research has come from the National Science and Engineering Research Council (NSERC Discovery Grant to L. Brown), the Queen Elizabeth II Graduate Scholarship in Science and Technology (QEII-GSST), and the Graduate Expansion Fund (GEF) from the University of Toronto Mississauga. We would also like to thank Stephen Howell and Michael Brady from Environment Canada for their valuable insight and technical support.

**Data**

All data sets used in this study are freely available online.

**Author Contributions**

Project conceptualization and methodology were designed by AD and LB, with the formal data analysis carried out by AD. Original draft was prepared by AD, with review and editing from both AD and LB.

**Competing Interests**

The authors declare that they have no conflict of interest

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

**Figure 1. Map of the study area, including the main Sea Ice, Snow and Lake Ice regions (coloured), and subregions (numbered) included in Table 3.**

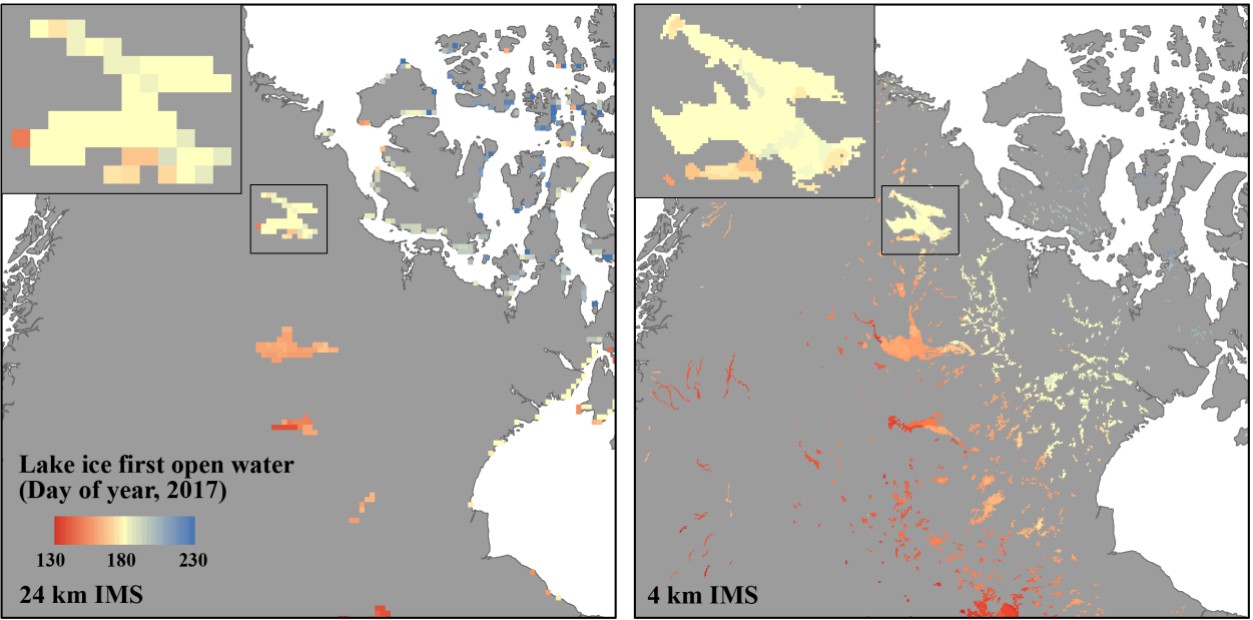


**Figure 2. Comparison of 24 km (left) and 4 km (right) lake ice first open water in 2017.**

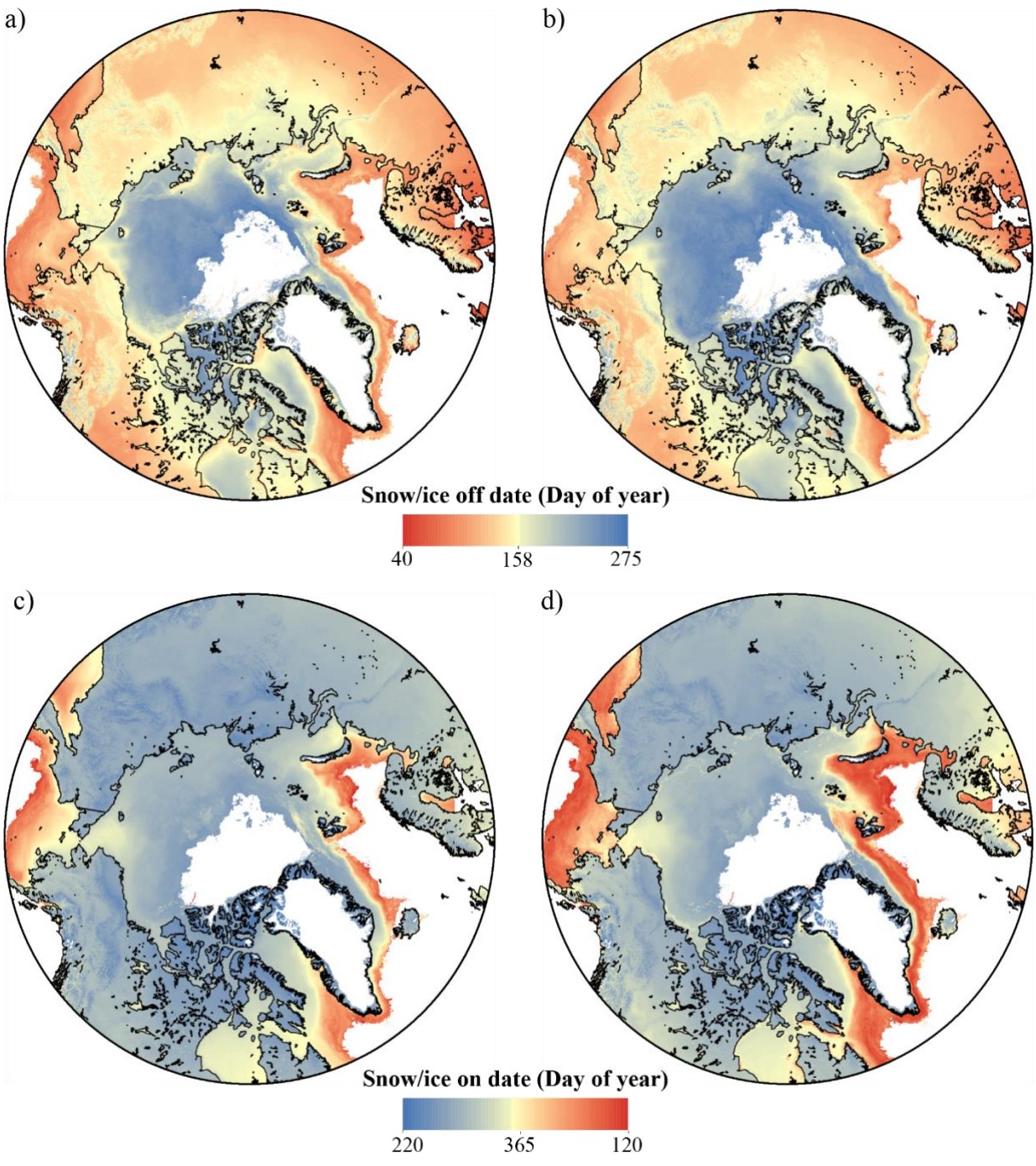

**Figure 3.** Mean 4 km IMS (2004 – 2019) (a) sea ice first open water (FOW$_S$), first snow-off (first_s$_{OFF}$), and lake ice first open water (FOW$_L$), (b) sea ice water clear of ice (WCI$_S$), final snow-off (final_s$_{OFF}$), and lake ice water clear of ice (WCI$_L$) , (c) sea ice freeze onset (FO$_S$), first snow-on (first_s$_{ON}$), and lake ice freeze onset (FO$_L$), and (d) sea ice continuous ice cover (CIC$_S$), final snow-on (final_s$_{ON}$), and lake ice continuous ice cover (CIC$_L$). White regions indicate where either no snow/ice forms, or snow/ice remains all year, in 14 or more years of the dataset.

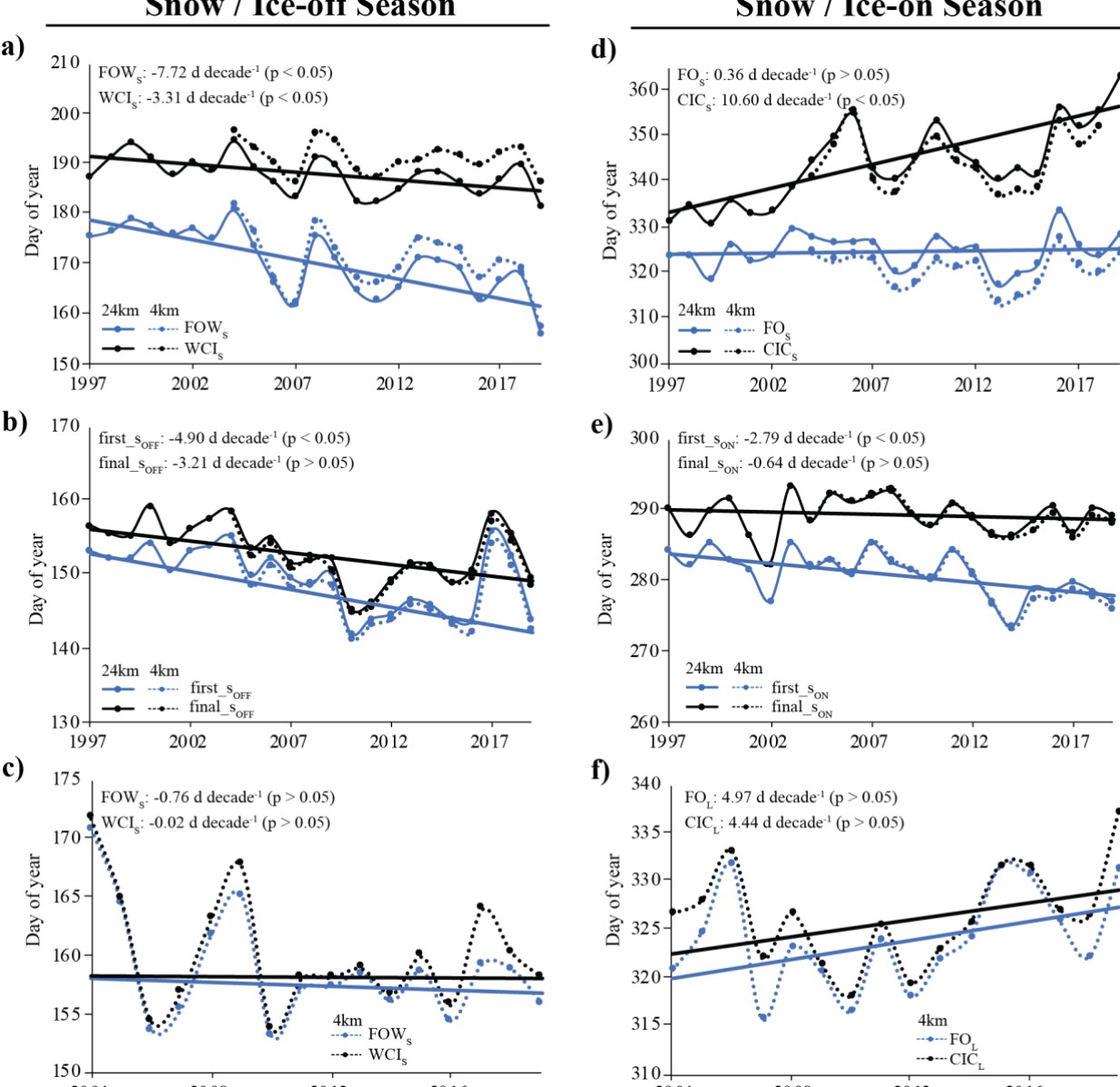

**Figure 4.** Mean 24 km (1997 – 2019) and 4 km (2004 – 2019) (a) IMS sea ice first open water (FOW_S) and water clear of ice (WCI_S) , (b) first snow-off (first_s_OFF) and final snow-off (final_s_OFF), (c) lake ice first open water (FOW_L) and water clear of ice (WCI_L), (d) sea ice freeze onset (FO_S) and continuous ice cover (CIC_S), (e) first snow-on (first_s_ON) and final snow-on (final_s_ON), and (f) lake ice freeze onset (FO_L) and continuous ice cover (CIC_L). Sen's slope and significance are indicated for each phenology parameter using the 24 km IMS product. Note that for lake ice, only the 4 km IMS product was used in this study.

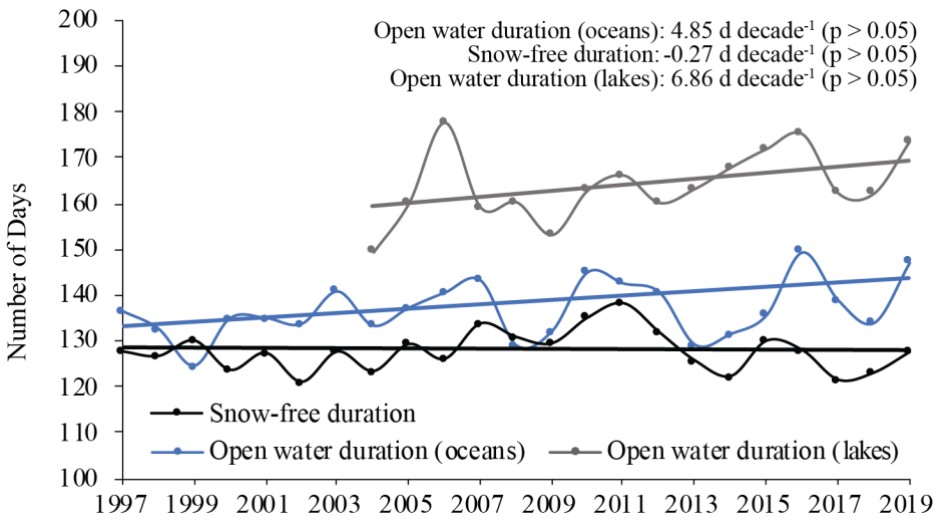


**Figure 5. Pan-Arctic open water duration for oceans (1997 – 2019), snow-free duration (1997 – 2019) over land, and open water duration for lakes (2004 – 2019). Sen's slope of the trend and significance are shown.**

**Snow / Ice-off Season**

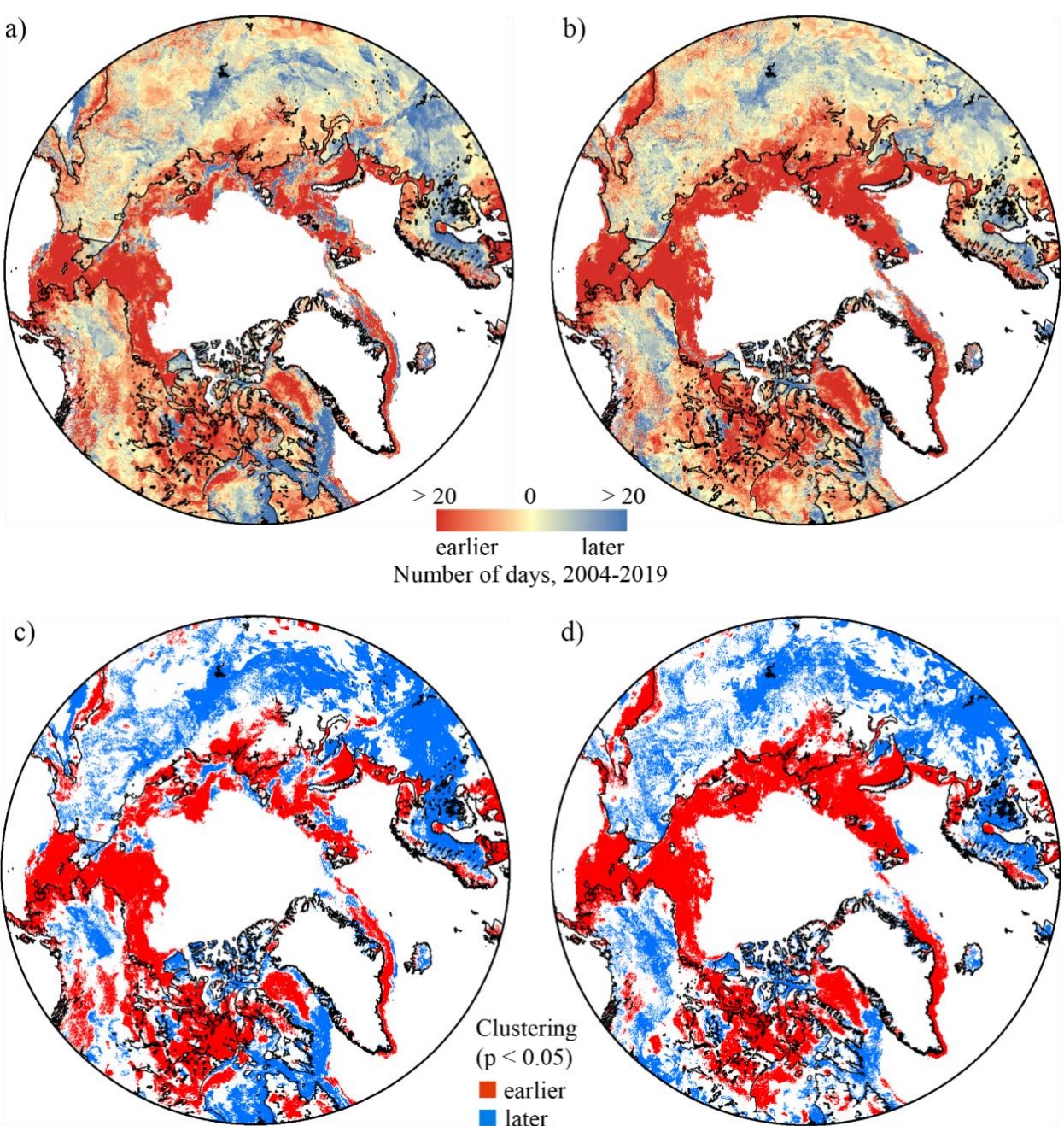

**Figure 6. Trends in 4 km IMS (2004 – 2019) (a) sea ice first open water (FOW$_S$), first snow-off (first_s$_{OFF}$), and lake ice first open water (FOW$_L$), (b) sea ice water clear of ice (WCI$_S$), final snow-off (final_s$_{OFF}$), and lake ice water clear of ice (WCI$_L$), (c) significant trend clusters in FOWS, FOW$_L$ and first_S$_{OFF}$, and (d) significant trend clusters in WCIS, WCI$_L$, and final_S$_{OFF}$.**

**Snow / Ice-on Season**

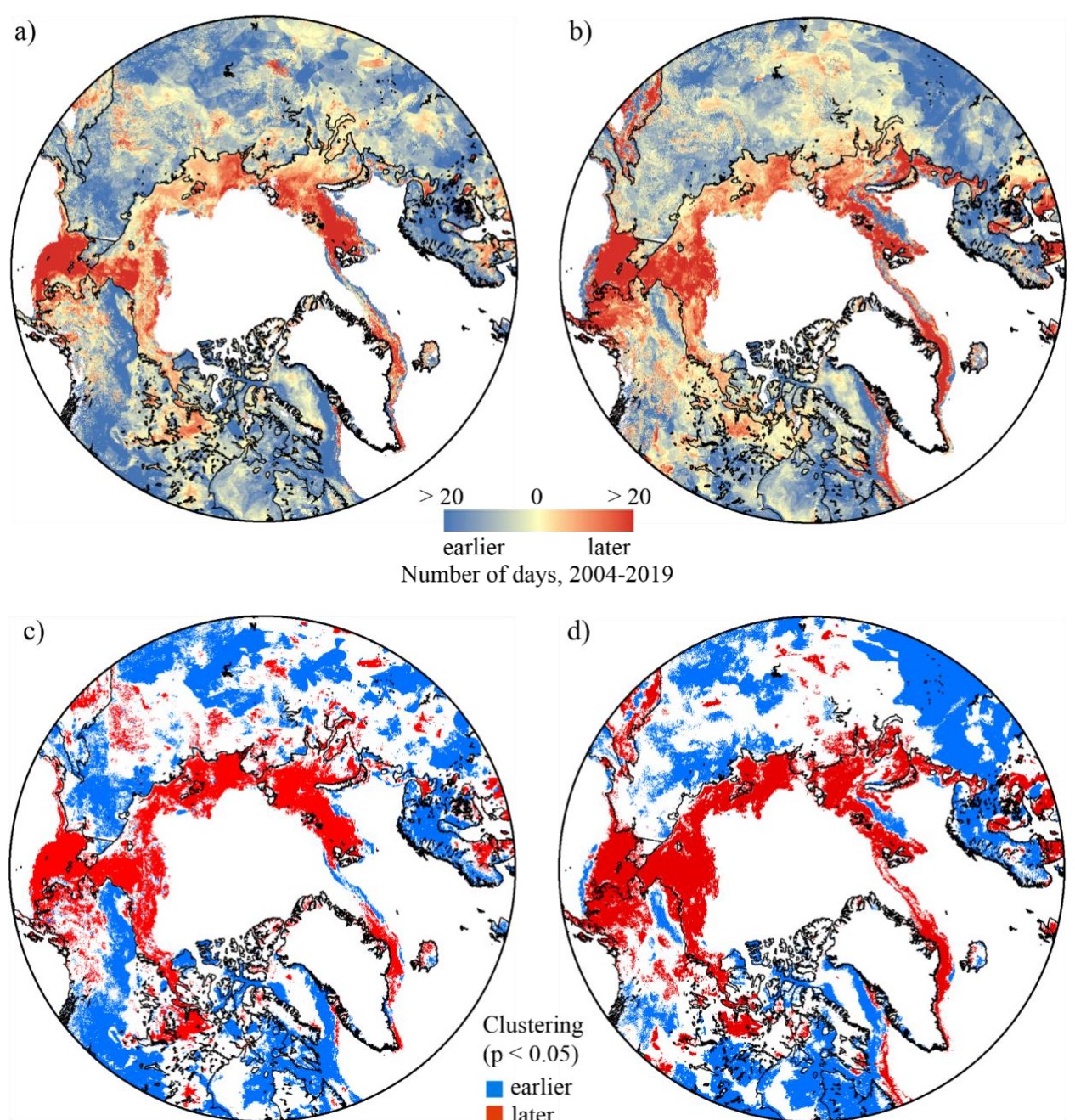

**Figure 7. Trends in 4 km IMS (2004 – 2019) (a) sea ice freeze onset (FO$_S$), first snow-on (first_s$_{ON}$), and lake ice freeze onset (FO$_L$), (b) and sea ice continuous ice cover (CIC$_S$), final snow-on (final_s$_{ON}$), and lake ice continuous ice cover (CIC$_L$), (c) significant trend clusters in FO$_S$, FO$_L$ and first_s$_{ON}$, and (d) significant trend clusters in CIC$_S$, CIC$_L$, and final_s$_{ON}$.**


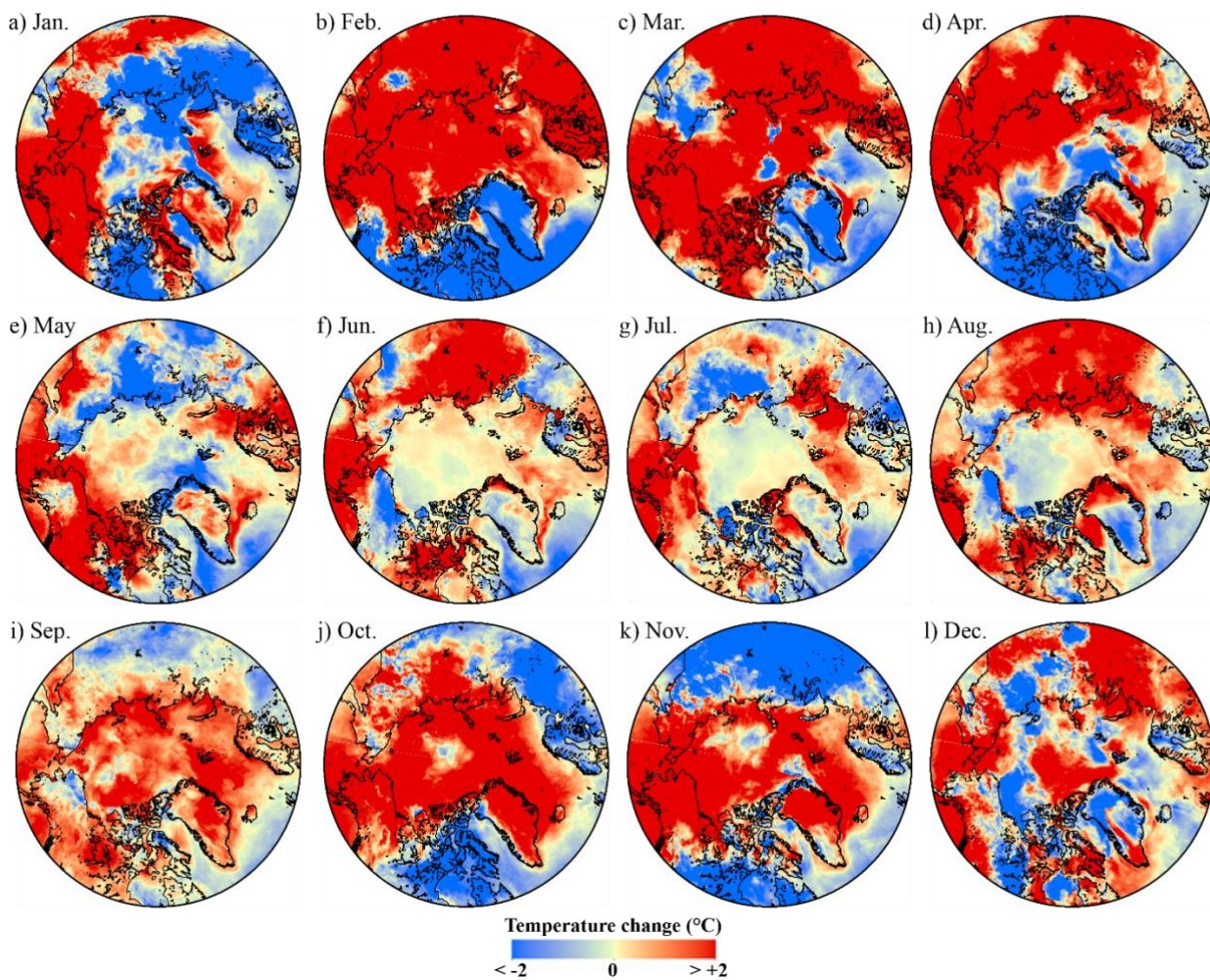


**Figure 8. Trends in monthly 2 m temperature from 2004 – 2019 in (a) January , (b) February, (c) March, (d) April € May, (f) June, (g) July, (h) August, (i) September, (j) October, (k) November, and (l) December.**


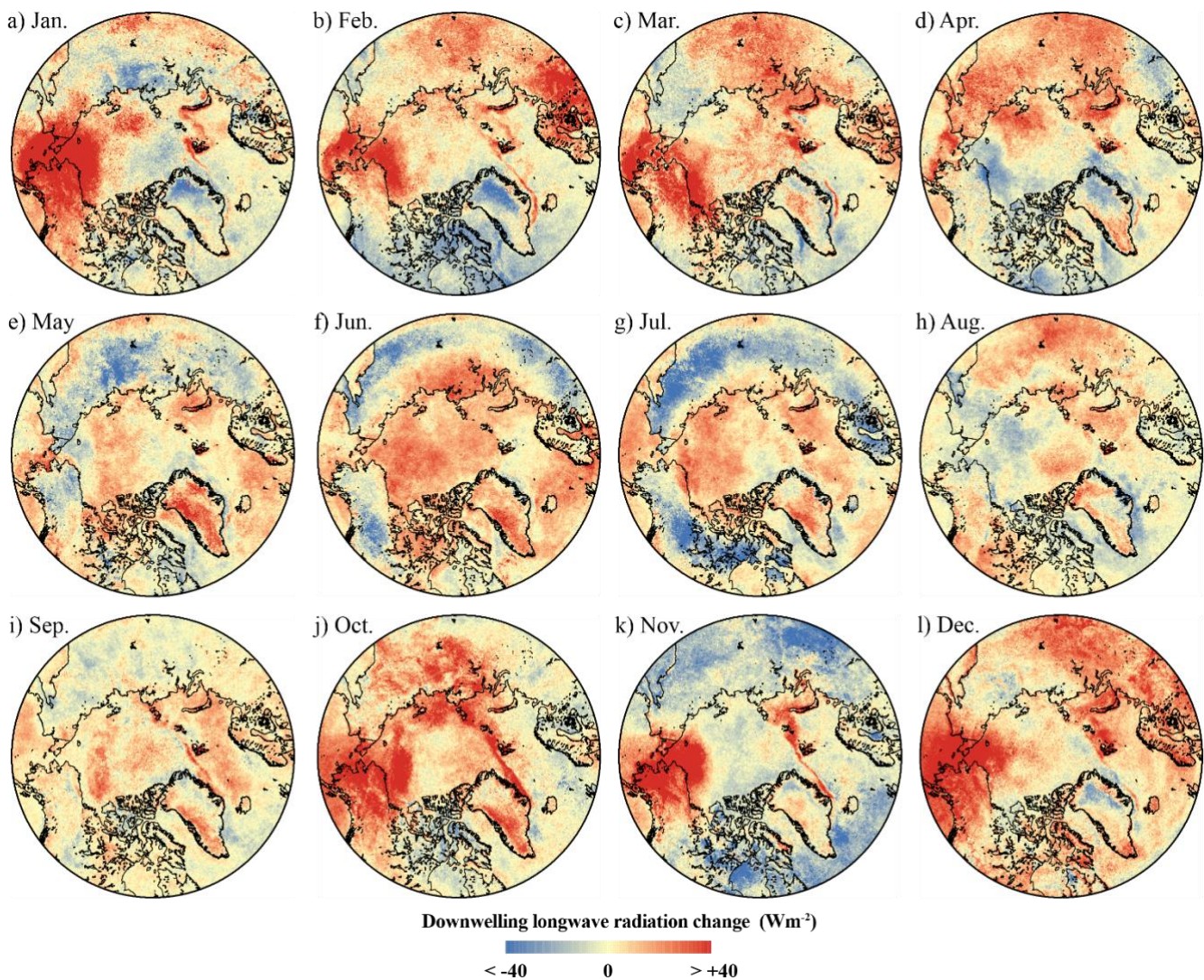

**Figure 9: –Trends in downwelling longwave radiation 2004 – 2019 in (a) January, (b) February, (c) March, (d) April € May, (f) June, (g) July, (h) August, (i) September, (j) October, (k) November, and (l) December.**

**Table 1. Sea ice, lake ice, and snow phenology parameters and definitions in this study.**

| Parameter | | Definition |
|---|---|---|
| $FOW_S$<br>$FOW_L$ | First open water (sea ice)<br>First open water (lake ice) | The first change from ice to water for a given pixel |
| $WCI_S$<br>$WCI_L$ | Water clear of ice (sea ice)<br>Water clear of ice (lake ice) | The last change from ice to water, signaling ice-free conditions for the remainder of the season |
| $FO_S$<br>$FO_L$ | Freeze onset (sea ice)<br>Freeze onset (lake ice) | The first detection of ice for a given pixel |
| $CIC_S$<br>$CIC_L$ | Continuous ice cover (sea ice)<br>Continuous ice cover (lake ice) | The date of the last change from water to ice |
| first_SOFF | First snow-off | The first change from snow-covered land to snow-free land for a given pixel |
| final_SOFF | Final snow-off | The last change from snow-covered to snow-free land, signalling snow-free conditions for the remainder of the season |
| first_SON | First snow-on | The first change from snow-free land to snow-covered land |
| final_SON | Final snow-on | The last change from snow-free to snow-covered land |

**Table 2. Pan-Arctic Spearman rank correlations (ρ) for snow and ice phenology dates using the 24 km (1997 – 2019) and 4 km (2004 – 2019) IMS products. \* represents statistically significant correlations at the 95% confidence level.**

| | *rho* ($\rho$) | |
|---|---|---|
| ***First melt*** | | |
| | first_$_{\text{SOFF}}$ | $FOW_L$ |
| $FOW_S$ | 0.38 (24 km) <br> 0.38 (4 km) | 0.62\* (4 km) |
| $FOW_L$ | 0.55\* (4 km) | - |
| ***Final melt*** | | |
| | final_$_{\text{SOFF}}$ | $WCI_L$ |
| $WCI_S$ | 0.46\* (24 km) <br> 0.64\* (4 km) | 0.72\* (4 km) |
| $WCI_L$ | 0.51\* (4 km) | - |
| ***First Freeze*** | | |
| | first_$_{\text{SON}}$ | $FO_L$ |
| $FO_S$ | 0.15 (24 km) <br> 0.08 (4 km) | 0.23 (4 km) |
| $FO_L$ | -0.27 (4 km) | - |
| ***Final Freeze*** | | |
| | final_$_{\text{SON}}$ | $CIC_L$ |
| $CIC_S$ | 0.37 (24 km) <br> 0.24 (4 km) | 0.37 (4 km) |
| $CIC_L$ | 0.19 (4 km) | - |

**Table 3. Regional analysis (see Figure 1) of the median trend strength (days/16years) and direction (- earlier, + later) for all of the phenology parameters: First open water (FOW, subscript S denotes sea ice, L denotes Lake ice), Water clear of ice (WCI, subscript S denotes sea ice, L denotes Lake ice), Freeze onset (FO, subscript S denotes sea ice, L denotes Lake ice), Complete ice cover (CIC, subscript S denotes sea ice, L denotes Lake ice), first and final snow-off (_$S_{OFF}$), first and final snow-on (_$S_{ON}$).**

| Sea Ice | Melt | | Freeze | |
|---|---|---|---|---|
| | $FOW_S$ | $WCI_S$ | $FO_S$ | $CIC_S$ |
| **Canadian Arctic Region** | **+2** | **-7** | **-11** | **-9** |
| Canadian Arctic Archipelago | -4 | -7 | -12 | -8 |
| Hudson Bay | +2 | -7 | -10 | -10 |
| Baffin Bay | -9 | -18 | -7 | -6 |
| Davis Strait | +24 | +11 | -22 | -10 |
| **Alaska/ Far East Russia Region** | **-23** | **-31** | **+8** | **+14** |
| Beaufort Sea | -30 | -37 | +6 | +8 |
| Chukchi Sea | -25 | -31 | +8 | +19 |
| Bering Sea | -34 | -41 | +27 | +52 |
| **Eurasia Region** | **-14** | **-28** | **+7** | **+10** |
| Eastern Siberian Sea | -15 | -24 | +5 | +7 |
| Laptev Sea | -11 | -28 | +8 | +8 |
| Kara Sea | -16 | -32 | +7 | +10 |
| Barents Sea | -15 | -34 | +16 | +13 |
| Greenland Sea | -13 | -25 | +2 | +15 |

| Snow / Lake Ice | Melt | | | | Freeze | | | |
|---|---|---|---|---|---|---|---|---|
| | First_$S_{OFF}$ | Final_$S_{OFF}$ | $FOW_L$ | $WCI_L$ | First_$S_{ON}$ | Final_$S_{ON}$ | $FO_L$ | $CIC_L$ |
| **North American Arctic** | **-8** | **-6** | **-4** | **-4** | **-8** | **-3** | **+2** | **0** |
| Canada Mainland West | -11 | -10 | -5 | -5 | -5 | -4 | +5 | +1 |
| Northern QC | -7 | -5 | +9 | +9 | -8 | -16 | -4 | -2 |
| Alaska/ Far East Russia | -3 | 0 | -18 | -19 | -5 | +1 | +9 | +11 |
| Western Alaska | -17 | -15 | -22 | -27 | +3 | +22 | +33 | +25 |
| North Slope | -8 | -1 | +3 | +3 | -21 | 0 | -8 | -9 |
| Far East Russia | 0 | 1 | NA | NA | -9 | -4 | NA | NA |
| Nettlling Lake | | | -3 | +2 | | | -3 | -1 |
| Amadjuak Lake | | | -1 | 0 | | | +3 | +2 |
| Great Slave Lake | | | -6 | -4 | | | +3 | +1 |
| Great Bear Lake | | | -4 | -8 | | | +8 | +6 |
| Lake Hazen | | | -1 | -4 | | | +3 | -11 |
| **Eurasia** | **0** | **0** | **-1** | **-2** | **-9** | **-7** | **+8** | **+8** |
| Scandinavia/Northern Europe | 0 | 0 | -1 | -1 | -13 | -9 | +28 | +19 |
| NW Eurasia | +5 | +3 | +1 | -2 | -8 | -13 | -4 | -6 |
| Central Eurasia | 0 | 0 | -7 | -9 | -9 | -4 | -2 | -4 |
| NE Eurasia | -1 | -1 | +2 | +2 | -9 | -8 | -6 | -9 |
| Lake Ladoga* | | | N/A | -9 | | | +13 | N/A |
| Lake Onega | | | -5 | -6 | | | +28 | +15 |

* $FOW_L$ and $CIC_L$ are not included for Lake Ladoga as the lake did not fully freeze in several of the study years.

**Table 4. Regional Spearman rank correlations (ρ) for snow and ice phenology dates and monthly 2 m temperature from 2004 - 2019 using 4 km IMS. For sea ice, 'Canadian Arctic' includes Baffin Bay, Hudson Bay, and the CAA; 'Alaska/Far East Russia' includes Beaufort, Chukchi, and Bering seas; 'Eurasian Arctic' includes East Siberian, Laptev, Kara, Bering, and Greenland seas (See Figure 1). Months were selected for each phenology parameter based on mean phenology dates in Figure 3. bold represents statistically significant correlations at the 95% confidence level.**

**Sea Ice**

| Melt | Canadian Arctic | | Alaska/Russia Arctic | | Eurasian Arctic | |
|---|---|---|---|---|---|---|
| | $FOW_S$ | $WCI_S$ | $FOW_S$ | $WCI_S$ | $FOW_S$ | $WCI_S$ |
| Mar | -0.174 | -0.309 | -0.356 | **-0.524** | 0.059 | -0.024 |
| Apr | -0.344 | -0.315 | 0.103 | -0.038 | **-0.579** | **-0.741** |
| May | -0.471 | **-0.562** | 0.185 | 0.488 | -0.135 | -0.068 |
| Jun | -0.441 | -0.482 | -0.165 | -0.006 | -0.209 | -0.300 |
| Jul | **-0.553** | **-0.653** | **-0.632** | -0.435 | **0.529** | 0.118 |
| Aug | **-0.668** | **-0.591** | -0.468 | -0.224 | 0.247 | 0.262 |
| Sep | **-0.747** | **-0.579** | **0.518** | **0.712** | 0.026 | 0.118 |

| Freeze | Canadian Arctic | | Alaska/Russia Arctic | | Eurasian Arctic | |
|---|---|---|---|---|---|---|
| | $FO_S$ | $CIC_S$ | $FO_S$ | $CIC_S$ | $FO_S$ | $CIC_S$ |
| Sep | 0.447 | **0.553** | -0.300 | -0.165 | 0.318 | -0.059 |
| Oct | 0.141 | 0.306 | -0.179 | 0.144 | 0.462 | 0.285 |
| Nov | 0.491 | **0.582** | 0.050 | 0.062 | 0.271 | 0.026 |
| Dec | 0.341 | 0.415 | 0.094 | -0.206 | **0.524** | 0.047 |
| Jan | 0.124 | -0.029 | -0.171 | 0.115 | 0.124 | 0.024 |
| Feb | 0.026 | 0.035 | 0.165 | 0.226 | 0.376 | 0.047 |
| Mar | **0.594** | 0.224 | 0.429 | 0.215 | -0.024 | -0.426 |
| Apr | 0.238 | 0.038 | -0.047 | -0.462 | -0.288 | 0.015 |

**Snow and Lake Ice**

| Melt | Eurasia | | | | North America | | | |
|---|---|---|---|---|---|---|---|---|
| | $first\_S_{off}$ | $final\_S_{off}$ | $FOW_L$ | $WCI_L$ | $first\_S_{off}$ | $final\_S_{off}$ | $FOW_L$ | $WCI_L$ |
| Mar | 0.050 | 0.124 | -0.406 | -0.359 | - | - | - | - |
| Apr | -0.235 | **-0.629** | **-0.665** | **-0.665** | **-0.618** | -0.382 | -0.409 | -0.400 |
| May | -0.338 | **-0.691** | -0.194 | -0.365 | **-0.612** | **-0.603** | **-0.506** | **-0.532** |
| Jun | 0.156 | -0.226 | -0.109 | -0.176 | -0.432 | -0.241 | **-0.615** | **-0.638** |
| Jul | -0.288 | **-0.500** | -0.203 | -0.326 | -0.321 | -0.259 | -0.485 | -0.488 |

| Freeze | Eurasia | | | | North America | | | |
|---|---|---|---|---|---|---|---|---|
| | $first\_S_{on}$ | $final\_S_{on}$ | $FO_L$ | $CIC_L$ | $first\_S_{on}$ | $final\_S_{on}$ | $FO_L$ | $CIC_L$ |
| Aug | -0.046 | 0.121 | - | - | **0.639** | 0.336 | - | - |
| Sep | -0.096 | -0.421 | 0.309 | 0.209 | **0.557** | 0.046 | 0.118 | -0.088 |
| Oct | 0.246 | -0.111 | -0.229 | -0.156 | **0.543** | 0.407 | 0.088 | 0.100 |
| Nov | 0.114 | 0.296 | 0.309 | 0.212 | -0.193 | -0.136 | **0.565** | **0.509** |
| Dec | 0.093 | 0.132 | 0.426 | 0.332 | 0.071 | -0.064 | 0.138 | 0.103 |
| Jan | - | -0.257 | -0.076 | -0.100 | - | - | 0.165 | 0.279 |
| Feb | - | -0.107 | 0.432 | **0.526** | - | - | - | - |

**Table 5.** Regional Spearman rank correlations (ρ) for snow and ice phenology dates and monthly downwelling longwave radiation (APP-x) from 2004 - 2019. For sea ice, 'Canadian Arctic' includes Baffin Bay, Hudson Bay, and the CAA; 'Alaska/Far East Russia' includes Beaufort, Chukchi, and Bering seas; 'Eurasian Arctic' includes East Siberian, Laptev, Kara, Bering, and Greenland seas (See Figure 1). Months were selected for each phenology parameter based on mean phenology dates in Figure 3. bold represents statistically significant correlations at the 95% confidence level.

**Sea Ice**

| Melt | Canadian Arctic | | Alaska/Russia Arctic | | Eurasian Arctic | |
|---|---|---|---|---|---|---|
| | $FOW_S$ | $WCI_S$ | $FOW_S$ | $WCI_S$ | $FOW_S$ | $WCI_S$ |
| Mar | 0.385 | 0.338 | **-0.659** | -0.456 | 0.156 | 0.106 |
| Apr | 0.021 | 0.103 | -0.124 | -0.141 | -0.259 | -0.326 |
| May | -0.244 | -0.253 | **-0.526** | -0.391 | 0.100 | -0.338 |
| Jun | -0.079 | -0.100 | -0.171 | -0.032 | 0.253 | -0.006 |
| Jul | 0.012 | -0.115 | -0.235 | -0.009 | -0.003 | -0.274 |
| Aug | -0.162 | -0.182 | -0.171 | 0.147 | 0.097 | -0.091 |
| Sep | -0.168 | **-0.497** | 0.144 | 0.450 | -0.215 | -0.406 |

| Freeze | Canadian Arctic | | Alaska/Russia Arctic | | Eurasian Arctic | |
|---|---|---|---|---|---|---|
| | $FO_S$ | $CIC_S$ | $FO_S$ | $CIC_S$ | $FO_S$ | $CIC_S$ |
| Sep | **0.512** | 0.241 | -0.068 | -0.238 | -0.056 | -0.209 |
| Oct | **0.585** | 0.379 | 0.053 | -0.156 | **0.506** | 0.171 |
| Nov | 0.485 | 0.306 | -0.197 | 0.082 | 0.221 | -0.079 |
| Dec | 0.412 | **0.671** | 0.229 | 0.456 | 0.256 | -0.044 |
| Jan | 0.300 | 0.218 | 0.265 | **0.665** | 0.462 | 0.129 |
| Feb | 0.268 | 0.409 | 0.279 | **0.594** | **0.506** | -0.065 |
| Mar | 0.232 | 0.191 | 0.468 | **0.562** | 0.347 | -0.118 |
| Apr | 0.344 | 0.385 | 0.185 | -0.056 | 0.224 | 0.009 |

**Snow and Lake Ice**

| Melt | Eurasia | | | | North America | | | |
|---|---|---|---|---|---|---|---|---|
| | $first\_S_{off}$ | $final\_S_{off}$ | $FOW_L$ | $WCI_L$ | $first\_S_{off}$ | $final\_S_{off}$ | $FOW_L$ | $WCI_L$ |
| Mar | 0.112 | 0.379 | -0.171 | -0.185 | - | - | - | - |
| Apr | 0.003 | **-0.318** | -0.300 | -0.447 | **-0.647** | -0.415 | -0.535 | -0.538 |
| May | -0.421 | -0.629 | -0.232 | -0.282 | **-0.529** | **-0.515** | -0.556 | **-0.565** |
| Jun | -0.182 | -0.374 | 0.191 | 0.038 | -0.156 | -0.106 | **-0.044** | **-0.082** |
| Jul | -0.279 | **-0.447** | -0.297 | -0.382 | -0.238 | -0.174 | **-0.406** | **-0.403** |

| Freeze | Eurasia | | | | North America | | | |
|---|---|---|---|---|---|---|---|---|
| | $first\_S_{on}$ | $final\_S_{on}$ | $FO_L$ | $CIC_L$ | $first\_S_{on}$ | $final\_S_{on}$ | $FO_L$ | $CIC_L$ |
| Aug | 0.232 | -0.021 | - | - | 0.264 | 0.318 | - | - |
| Sep | 0.346 | -0.321 | 0.259 | 0.221 | **0.568** | 0.118 | 0.271 | 0.024 |
| Oct | 0.236 | -0.082 | 0.047 | 0.106 | 0.264 | 0.332 | -0.135 | -0.162 |
| Nov | -0.250 | 0.232 | 0.188 | 0.309 | -0.243 | -0.229 | 0.247 | 0.071 |
| Dec | -0.236 | -0.075 | 0.188 | 0.091 | -0.346 | -0.286 | 0.071 | -0.018 |
| Jan | - | -0.446 | 0.188 | 0.191 | - | - | 0.218 | 0.306 |
| Feb | - | -0.261 | 0.238 | 0.353 | - | - | - | - |

935

940