# Peer review of "Recent changes in Pan-Arctic sea ice, lake ice, and snow on/off timing"

_The Cryosphere, 2021_

## Referee Comment (RC1)

**Major Concerns**
This study looks at relationships between sea ice, snow cover and lake ice timings of retreat/advance from 1997 - 2019 (shorter time-period for lake ice). This is done on a pan-Arctic scale first and then more regional discussions follow. While some relationships between exist based on large-scale warming, earlier snow cover may be expected because of increased atmospheric water vapor from more open water in autumn. Studies have documented that the Arctic atmosphere is now warmer and wetter (i.e. Boisvert and Stroeve, 2015; Serreze et al. 2012), and links with earlier snow on land have been reported (i.e. Ghatak et al. 2012). However, none of these studies are referenced here in the discussion of these linkages, and all relationships are really just discussed in terms of air temperature, which is overly simplistic. Drivers for example for earlier melt onset over sea ice is largely driven by warm/moist air advection into the Arctic (i.e. see papers by Kapsch, Francis, Mortin), especially on the Eurasian side of the Arctic.  And thus downwelling longwave has been found to be the primary driver. And for freeze-up, Steele et al. talked about the importance of mixed layer ocean heat driving the freeze-up and of course many papers have then discussed this heat release to the atmosphere driving the warming air temperatures (i.e. papers by Serreze et al., Screen et al., etc). Thus the maps, while interesting, are analyzed in terms of their inter-relationships only with correlations and air temperature. You may not be aware of a publication by Crawford et al. (2018: JGR-atmosphere) who looked at the relationship between snow retreat and sea ice retreat. The only region with a statistically significant relationship was found in the Laptev Sea with the early retreat of snow over the West Siberian Plain. They discussed the atmospheric mechanisms by which these regions could be linked. Such an analysis could be applied here in a broader context to really understand how these components, sea ice, lake ice and snow cover are interconnected. Just because you have a correlation between two variables doesn't mean you've understood the drivers of the linkages.

So then I have to ask what are we gaining from this study? The conclusions state that there are trends towards a longer snow and ice-free season, which isn't new knowledge, right? But it is interesting that the ice-free or snow-free season is largest for the sea ice, followed by the lake ice and then the snow cover. So what are the implications of that? And why would sea ice have a longer ice-free season? These are some of the questions that should be answered by this study. Otherwise it just feels like a missed opportunity.

**Other Comments**
As far as I know, IMS data have never been recommended for use in long-term change studies because of the dependence on analysts interpreting if there is ice or snow there or not. Analysts change and there is always a subjective element to this mapping. How are you accounting for error in your assessment?  I would think at minimum checking your results at least for sea ice and snow cover against automated data products would be worthwhile. And some discussion on how your ice/snow on/off dates relate to earlier studies is needed. I realize that your time-period will differ from other studies,

I think it's good to discuss a bit more about your methods for first date of no ice and date when the ice is gone for good until it comes back. Obviously this is a bit problematic for sea ice which is in constant motion. Stammerjohn et al. 2012 and Stroeve et al. 2016 had different ways of computing the retreat/advance of sea ice. This will also play into your determination of the length of the open water period. And of course this influences your trends shown for first ice-off and continuous ice off dates, and why they differ so much between first open water/continuous open water and first freeze/continuous freeze.

It is not particularly novel to discuss sea ice retreat/advance or snow off/on as this has been done already in other studies. However, I do like seeing Figures 3 -5 as it's nice to see the land and ocean at the same time. However, talking about mean values for the Arctic as a whole is really meaningless and you rightly point out that there are large regional differences. Best to focus on lags between the ocean and land regions on a regional basis. Thus, in general I think the focus really should be on the regional relationships but then some sort of advanced clustering analysis would be beneficial to first identity the regions with the strongest relationships. Crawford et al. (2018) did show some ways to do this analysis that could be useful here.

Line 45: I find this statement to be a bit strange since timing of melt onset, and melt onset trends are latitudinal dependent, so I would expect the CAA to have weaker trends in melt onset and freeze-up. For the most up-to-date trends in melt onst/freeze-up you could reference Stroeve and Notz, 2018.

Line 66: on the other hand, once there is liquid water in the snow and/or melt water, passive microwave algorithms underestimate sea ice area, so are you just referring to coastal areas here? Yes there is coastal contamination, resulting in false ice concentrations near the coast, yet in summer you also have the opposing effect of melt water.

Line 181: I do not understand your statement that sea ice off dates are most strongly correlated with temperature in September. I assume you're speaking of air temperature so you should specify this. Yet it is well known that air temperature does not drive melt onset or ice retreat (i.e. Mortin et al., 2014; Kapsch et al. ). Basically what drives earlier melt onset (which is correlated with ice retreat – Stroeve et al., 2016) is advection of warm moist air masses into the Arctic and the downwelling longwave and sensible/latent heat fluxes associated with those air masses. This is especially true in the eastern Arctic (i.e. Barents and Kara seas). Maybe in the CAA air temperatures are important but this is not true everywhere in the Arctic. Also, why would you expect to have the strongest correlation with September temperatures? Unless it's a feedback that earlier retreat of open water leads to warmer ocean mixed layer temperatures that then of course help keep the atmospheric temperatures warmer. Nevertheless, I think your analysis here is too simplistic and doesn't add anything unless you look at all drivers that influence ice/snow retreat/advance.  Thus, the same concerns will apply later on lines 196:200 since if you have earlier retreat of ice you will have later ice advance (i.e. Stroeve et al., 2016; Steele et al. 2016), and of course the release of the mixed layer heat back to the atmosphere

will be responsible for the correlation you see (see also papers about Arctic Amplification by Serreze et al. 2009; Screen et al. )

Line 200: I believe your results are consistent with several studies indicating earlier snow fall in autumn in part because of the sea ice loss. This paper comes to mind (Ghatak et al. 2010 – JGR) but I believe Judah Cohen has also written on this.

Line 233: How is the clustering done? This wasn't discussed in the methodology section.

Lines 423-424: I disagree that delays in freeze-up are consistent with delayed snow onset over land and delayed freeze-onset on lakes. Yes, the lack of sea ice may result in locally warmer air temperatures that influence snow and lake ice formation, yet the lack of sea ice may also lead to earlier autumn snow accumulation. I feel your simple statistical analysis does not really explain the processes for the relationships observed.

---

## Author Comment (AC1)

We thank Reviewer #1 for the insightful suggestions that have greatly improved the manuscript.

Overall manuscript revisions:

The correlation and trend analysis were repeated for the 925 mb temperatures (ERA5) and downwelling longwave radiation at the surface (APP-x), though only the longwave radiation is included in the final version of the manuscript.

Trend sentences are all group together now, followed by the correlations between the parameters at the pan arctic scale. Correlations between pan-arctic phenology and air temperature / downwelling longwave radiation are focussed on in the regional section. Additional information regarding the phenology timing has been added to each region to enhance the discussion and provide context for the correlations. All correlation numbers have been removed to improve readability, and only trend number remain.

All comments have been responded to below, and line numbers below refer to the tracked changes document with all revisions showing.

**Major Concerns**

This study looks at relationships between sea ice, snow cover and lake ice timings of retreat/advance from 1997 - 2019 (shorter time-period for lake ice). This is done on a pan-Arctic scale first and then more regional discussions follow. While some relationships between exist based on large-scale warming, earlier snow cover may be expected because of increased atmospheric water vapor from more open water in autumn. Studies have documented that the Arctic atmosphere is now warmer and wetter (i.e. Boisvert and Stroeve, 2015; Serreze et al. 2012), and links with earlier snow on land have been reported (i.e. Ghatak et al. 2012).

We used a citation to Thackery et al., 2019 to explain the warmer wetter atmosphere where relevant in the text, but have now revised to include Boisvert and Stroeve, 2015 as well and to mention Ghatak et al 2010 and 2012 with respect to increased snow fall in the region where we identify earlier trends. Boisvert and Stroeve, 2015, is used in particular now to explain the correlations between ice-on and downwelling longwave radiation at the surface. In so much as the correlation is not saying that freeze is triggered by downwelling longwave, but the feedbacks related to longer open water and increased cloud cover are resulting in increased downwelling longwave radiation during the freeze season.

However, none of these studies are referenced here in the discussion of these linkages, and all relationships are really just discussed in terms of air temperature, which is overly simplistic. Drivers for example for earlier melt onset over sea ice is largely driven by warm/moist air advection into the Arctic (i.e. see papers by Kapsch, Francis, Mortin), especially on the Eurasian side of the Arctic. And thus downwelling longwave has been found to be the primary driver.

We redid the analysis using downwelling Longwave as a parameter to compare to the phenologies – the results show some useful links, though likely mask the local connections since the region are so broad. Some interesting regional patterns in the trends are clear from the new Figure 9 added, so we have added this new analysis throughout the manuscript as it adds some interesting components to the discussion.

Lines 161-174: Downwelling longwave radiation has been linked to melt onset in the Arctic Ocean (e.g., Mortin et al., 2016). To further explore the linkages in the phenology data, downwelling longwave radiation data from the Extended AVHRR Polar Pathfinder (APP-x) was obtained from NOAA National Centres for Environmental Information (https://www.ncei.noaa.gov/data/avhrr-polar-pathfinder-extended/access/)(Key et al., 2019). APP-x data is provided as 25 km EASE grid projection, processed for 0400 and 1400 (LST). Due to large areas of missing data between ~ 59 - 64°N, the mean monthly values were created from the 0400 and 1400 separately to avoid averaging errors where data exist for one time and not the other (to avoid skewing the average with the diurnal differences). Some artificial patterns are evident in the data (e.g., March, Figure 9c, near the pole), however for the purpose of regional comparisons this is not limiting as this region is not used in quantitative comparisons. Downwelling longwave radiation at the surface is calculated using a neural network to simulate a radiative model (see Key and Schweiger 1998; Key et al., 2016). Downwelling longwave radiation was selected from APP-x rather than ERA5 as the APP-x dataset has been determined as 'climate data record quality' and has been validated against in situ data with a bias of only 2.1 Wm-2 and RMSE of 22.4 Wm-2 (with the higher RMSE values attributed to differences in surface snow fall between the sampling site and the 25 km x 25 km area represented) (Key et al, 2016).

[Figure]

**Figure 9: –Trends in downwelling longwave radiation 2004 – 2019 in (a) January , (b) February, (c) March, (d) April € May, (f) June, (g) July, (h) August, (i) September, (j) October, (k) November, and (l) December.**

And for freeze-up, Steele et al. talked about the importance of mixed layer ocean heat driving the freeze-up and of course many papers have then discussed this heat release to the atmosphere driving the warming air temperatures (i.e. papers by Serreze et al., Screen et al., etc). Thus the maps, while interesting, are analyzed in terms of their inter-relationships only with correlations and air temperature. You may not be aware of a publication by Crawford et al. (2018: JGR-atmosphere) who looked at the relationship between snow retreat and sea ice retreat. The only region with a statistically significant relationship was found in the Laptev Sea with the early retreat of snow over the West Siberian Plain. They discussed the atmospheric mechanisms by which these regions could be linked. Such an analysis could be applied here in a broader context to really understand how these components, sea ice, lake ice and snow cover are interconnected. Just because you have a correlation between two variables doesn't mean you've understood the drivers of the linkages.

Our linkages are in the LISA maps – showing where the sea ice and snow/lake ice are changing similarly. See expanded comment below.

So then I have to ask what are we gaining from this study? The conclusions state that there are trends towards a longer snow and ice-free season, which isn't new knowledge, right? But it is interesting that the ice-free or snow-free season is largest for the sea ice, followed by the lake ice and then the snow cover. So what are the implications of that? And why would sea ice have a longer ice-free season? These are some of the questions that should be answered by this study. Otherwise it just feels like a missed opportunity.

Revised the conclusions to explicitly state the arctic amplification is stronger over the water than land and provide examples from the lakes:

Lines 615-627: … In Eurasia the snow cover trends are stronger in the freeze season, and predominantly towards earlier snow-on, while later lake freeze is occurring on the large lakes. Earlier snowfall occurs through this region and is related to feedbacks from the longer open ocean water, however the contrast in trends between snow and lake ice here show that the heat retained in the mixed layer of the lakes through the longer open water season is enough to delay freeze, despite snow falling earlier in the fall/winter. Overall, stronger trends towards longer open water duration on both the northern oceans and lakes are shown compared to the lack of overall trend in snow-free duration (the earlier snow-off trends are offset by the earlier snow-on trends) (Figure 5). This is in line with stronger Arctic Amplification processes over the Arctic Ocean compared to land (e.g., Miller et al., 2010), with the lower albedo of water allowing for more energy absorption and increased heating than occurs on land. This would apply to lakes as well and is particularly evident in lakes through Alaska with stronger trends towards earlier ice-off and later ice-on compared to snow, as well as in Scandinavia/Northern Europe where strong opposite trends are shown between later lake ice-on and earlier snow-on. Furthermore, feedbacks related to ocean-atmosphere interactions during the longer open water season are contributing to earlier snow-on timing in some regions.

**Other Comments**

As far as I know, IMS data have never been recommended for use in long-term change studies because of the dependence on analysts interpreting if there is ice or snow there or not. Analysts change and there is always a subjective element to this mapping. How are you accounting for error in your assessment? I would think at minimum checking your results at least for sea ice and snow cover against automated data products would be worthwhile. And some discussion on how your ice/snow on/off dates relate to earlier studies is needed. I realize that your time-period will differ from other studies,

More information from previous studies has been added now regarding IMS comparisons to sea ice, snow and lake ice.

Lines 128 – 141: IMS has been shown to outperform data from traditional passive microwave products (AMRS-E, SSM/I, SSMI-SSMIS) for both the timing and extent of first open water in the Arctic (Brown et al., 2014). For example, through the Barrow Strait in the CAA, the ability of the 4 km IMS data to resolve narrow channels lead to 17% more open water detected than with SSM/I, and 35% more open water than detected with AMSR-E, validated with RADARSAT-1 (Brown et al., 2014). Overall, most pixels compared between IMS and the two passive microwave datasets for first open water were within ± 5 days, with a greater percentage of the pixels in the categories beyond the ± 5 days identifying open water earlier with IMS than the other products (Brown et al., 2014). IMS has been shown to map higher snow cover fractions during the spring melt period than other snow products (Brown et al. 2010; Frei and Lee, 2010), but is reported to have mostly between 80-90% agreement with other snow products during the winter season of non-arctic North America, with better agreement in the later part of the winter season when deeper and more extensive snow cover is present (Chen et al., 2012). For lake ice, the 4 km IMS product occasionally identifies earlier lake ice-on dates in regions of prolonged cloud cover (e.g., northern Quebec, Canada), though both ice-on and -off timing detected using IMS are significantly correlated with, and comparable to, phenology dates extracted from the MODIS Snow Cover product (Brown and Duguay, 2012).

I think it's good to discuss a bit more about your methods for first date of no ice and date when the ice is gone for good until it comes back. Obviously this is a bit problematic for sea ice which is in constant motion. Stammerjohn et al. 2012 and Stroeve et al. 2016 had different ways of computing the retreat/advance of sea ice. This will also play into your determination of the length of the open water period. And of course this influences your trends shown for first ice- off and continuous ice off dates, and why they differ so much between first open water/continuous open water and first freeze/continuous freeze.

Table 1 explains how each phenology parameter was determined. This text was included in a table, rather than the manuscript to save space and make for an easier read. As noted, we did not track intermediate changes between ice/water and land/snow (though this is possible to do in future work, just not with how we configured the search algorithm for this project). More detail has been added including the definition of the open water and snow free timing:

Lines 179-186: Only the first and last change from ice/water and vice versa are tracked for this work, giving first and final dates of change. In sea ice regions dominated by thermodynamics, there is little difference between first and final timing, whereas in more active ice regions there

could be a more notable difference between the first and final timings as the ice moves past that pixel. Most lakes are dominated by thermodynamics and return similar first and final dates, however, lakes with more ice motion (e.g., Lake Onega and Ladoga) may show a difference in their timings.  For snow, warmer regions where more frequent snowmelt occurs tend to show a larger variation in first and final dates compared to the northern regions where the snow typically remains on the ground for the season. Open water duration and snow-free duration are defined as the time between the final change in the spring to the first change in the fall (WCI$_S$ to FO$_S$, and last_s$_{OFF}$ to first_s$_{ON}$).

It is not particularly novel to discuss sea ice retreat/advance or snow off/on as this has been done already in other studies. However, I do like seeing Figures 3 -5 as it's nice to see the land and ocean at the same time. However, talking about mean values for the Arctic as a whole is really meaningless and you rightly point out that there are large regional differences.

Agreed, we have removed the pan-arctic correlations, however, we chose to leave in the 24km and 4km pan-Arctic trends as they 'tell the story' of the large-scale phenology links before delving into the regional specifics.

Best to focus on lags between the ocean and land regions on a regional basis. Thus, in general I think the focus really should be on the regional relationships but then some sort of advanced clustering analysis would be beneficial to first identity the regions with the strongest relationships. Crawford et al. (2018) did show some ways to do this analysis that could be useful here.

Our cluster analysis uses Local indicators of spatial association (Anselin, 1995) and the maps show the areas if statistically significant spatial clusters. The pixels that show statistically significantly clustering were mapped into categories showing the clusters of either earlier or later trends, across the sea ice, snow and lake ice. Clusters crossing the shorelines indicate significantly clustered trends between the phenologies. The aim here was to see if the phenologies are changing similarly in geographic proximity. The LISA maps were not utilized well in the original manuscript and more discussion has been added throughout highlighting the regions where the ocean/land trends are clearly linked.  Crawford et al., 2019 use an interesting approach for comparisons, but a detailed regional analysis like this applied to our project is beyond the scope. Our results are focussed on comparing trends changing in the same season, rather than the effects across the season – which would be an excellent addition, but again, beyond the current scope. Our results show limited inland clustering of the snow and sea ice trends (towards earlier ice off and earlier final snow off) retreat in the Laptev Sea area focussed on in Crawford et al., 2018, but do not extend into sub regional correlations. We also show a difference in the snow off trend compared to Crawford et al. 2018.

Revised the methodology sentences to explicitly state how clusters that cross the shorelines indicate regions of spatially significant trends:

Lines 218-220: Clusters of spatially statistically significant trends of high and low trend strengths were mapped. Clusters crossing the shorelines indicate significantly clustered trends between the sea ice and snow or lake ice phenology parameters and show regions of interest where the phenology variables were responding with similar trend strength over the study period.

Line 45: I find this statement to be a bit strange since timing of melt onset, and melt onset trends are latitudinal dependent, so I would expect the CAA to have weaker trends in melt onset and freeze-up. For the most up-to-date trends in melt onst/freeze-up you could reference Stroeve and Notz, 2018.

Unclear what about this statement is strange – there are weaker melt season trends (for clarification - IMS is not detecting melt onset – it is detecting open water) in the CAA than elsewhere, and trends in freeze are towards earlier freeze up dates, see Dauginis and Brown, 2020. The CAA is in a region of less pronounced warming than elsewhere in the Arctic during the recent years and experiencing some cooling trends especially on the eastern side. The updated reference for the trend comparisons has been included now, thank you.

Line 66: on the other hand, once there is liquid water in the snow and/or melt water, passive microwave algorithms underestimate sea ice area, so are you just referring to coastal areas here? Yes there is coastal contamination, resulting in false ice concentrations near the coast, yet in summer you also have the opposing effect of melt water.

Yes, this was referring to the coast regions, for balance we've added:
Line 68-70:  … while in contrast it is known that that passive microwave data can underrepresent sea ice coverage when liquid water is present (melt ponds on the ice, or wet snow) (e.g. Meier, 2005).

Line 181: I do not understand your statement that sea ice off dates are most strongly correlated with temperature in September. I assume you're speaking of air temperature so you should specify this. Yet it is well known that air temperature does not drive melt onset or ice retreat (i.e. Mortin et al., 2014; Kapsch et al. ). Basically what drives earlier melt onset (which is correlated with ice retreat – Stroeve et al., 2016) is advection of warm moist air masses into the Arctic and the downwelling longwave and sensible/latent heat fluxes associated with those air masses. This is especially true in the eastern Arctic (i.e. Barents and Kara seas). Maybe in the CAA air temperatures are important but this is not true everywhere in the Arctic.

This was phrased poorly originally – the intention was that of the months used to explore freeze, September was the strongest correlation. Not that the strongest correlation over all possible parameters was September air temperature.  The entire correlation section has been re-written and now includes the downwelling longwave as well.

Also, why would you expect to have the strongest correlation with September temperatures? Unless it's a feedback that earlier retreat of open water leads to warmer ocean mixed layer temperatures that then of course help keep the atmospheric temperatures warmer. Nevertheless, I think your analysis here is too simplistic and doesn't add anything unless you look at all drivers that influence ice/snow retreat/advance. Thus, the same concerns will apply later on lines 196:200 since if you have earlier retreat of ice you will have later ice advance (i.e. Stroeve et al., 2016; Steele et al. 2016), and of course the release of the mixed layer heat back to the atmosphere will be responsible for the correlation you see (see also papers about Arctic Amplification by Serreze et al. 2009; Screen et al. )

Added some comments about the inherent feedbacks throughout the revised manuscript.

Line 200: I believe your results are consistent with several studies indicating earlier snow fall in autumn in part because of the sea ice loss. This paper comes to mind (Ghatak et al. 2010 – JGR) but I believe Judah Cohen has also written on this.

Thank you, these references have been added.

Line 233: How is the clustering done? This wasn't discussed in the methodology section.

The clustering is explained at the end of the methods section, local indicators of spatial association (LISA). We have expanded on this briefly, as mentioned above.

Lines 423-424: I disagree that delays in freeze-up are consistent with delayed snow onset over land and delayed freeze-onset on lakes. Yes, the lack of sea ice may result in locally warmer air temperatures that influence snow and lake ice formation, yet the lack of sea ice may also lead to earlier autumn snow accumulation. I feel your simple statistical analysis does not really explain the processes for the relationships observed.

This comment was related specifically to the Alaska region in the previous sentence where little to no increases in snow on timing are observed except for the swath of earlier snow through the North Slope. Revised the sentence to read:

Line 610: "Delays in sea ice freeze were also observed here (trends of 8 and 14 days later for first and final freeze) with much stronger trends in the Bering Sea region, along with delayed snow onset over land and delayed freeze onset in lakes across most of Alaska"

---

## Author Comment (AC2)

We thank Review 2 for the very helpful and constructive comments that we feel have greatly improved our manuscript. All comments have been addressed below (please see the blue coloured text). The line numbers below refer to the tracked changes document, with all revisions showing.

Overall manuscript revisions:

The correlation and trend analysis were repeated for the 925 mb temperatures (ERA5) and downwelling longwave radiation at the surface (APP-x), though only the longwave radiation is included in the final version of the manuscript.

Trend sentences are all group together now, followed by the correlations between the parameters at the pan arctic scale. Correlations between pan-arctic phenology and air temperature / downwelling longwave radiation are focussed on in the regional section. Additional information regarding the phenology timing has been added to each region to enhance the discussion and provide context for the correlations. All correlation numbers have been removed to improve readability, and only trend numbers remain.

**Summary**

This paper presents trends in the time of sea ice, snow, and lake ice presence based on the IMS product. Correlations are done with ERA5 2 m temperature data. The trends show earlier sea ice retreat and later freeze-up, in agreement with previous studies. The snow trends indicate earlier snow melt but also earlier first snow, so that trends in snow length are small. There is regional variability with the Bering and Chukchi Seas showing the greatest warming.

**General Comment:**

Overall, this is a competent manuscript. The data used are generally good (though see some comments below on this aspect) and the analysis is solid. However, it is lacking a coherent "story". The goal of the paper is to simultaneously highlight changes in sea ice, lake ice, and snow, but each is largely treated in parallel – as if there are three different papers just merged together. I'd like to see more discussion of connections and interactions, covariance in parameters, etc. For example, Figure 6 shows snow/ice-off trends and I see a distinct difference in Lake Ladoga versus the surrounding snow on land with the lake showing a much stronger trend toward earlier onset. I don't see a similar pattern in the Canadian lakes for snow/ice-off, but in Figure 7, some lakes – e.g., Great Bear Lake - do show a unique trend compared to the surrounding land. Digging into these types of things would be pretty interesting, but they aren't discussed in the paper. In short, based on the title and the abstract, I was expecting more of a co-variance analysis – how are lakes and nearby snow on land and sea ice correlated and how might the forcing mechanisms affect them similarly or differently.

The LISA maps were meant to indicate where the trends are significantly clustered between snow/sea ice/lake ice and highlight the links - but they were not discussed adequately in the manuscript. More sentences have been added highlighting similarities and differences in the trends

with some more detail, with extra focus on the large lakes. Lake specific comments added listed below. Other sentences on links are throughout the manuscript.

[revised manuscript text omitted]

The correlation with temperature is also rather superficial, simply noting correlation coefficients. I don't think the paper needs to be an in-depth attribution study. But simply noting correlation doesn't really provide much insight. I note below a couple examples where the paper could dig in a bit more – e.g., what is the cause of the correlations? The way it is presented, it suggests that higher temperatures result in earlier ice/snow loss and later ice/snow formation, but it's not necessarily that simple. For one thing obviously, there is more than just SAT going on – there are winds, SLP, ocean temperatures, sea ice advection. Just looking at SAT may not tell one much, especially in specific regions.

Downwelling longwave radiation has now also been added to the analysis to represent the combined effects of the temperature and moisture fluxes that influence sea ice citing literature highlighting the importance of downwelling longwave and comments rom Reviewer #1. Better links between the correlations and the ice information have been added throughout the manuscript to improve the discussion, though no further detailed analysis was done as the idea was present an overview more so than a detailed examination that would really need to be done an even more regional scale.

I also think that Section 3 is very wordy and somewhat hard to follow – it is one number after another. The data are all in the figures and tables, right? So, I don't think you need to worry so much about putting the numbers in the text. Focus on describing the main characteristics – significantly positive and negative trends, regional variations, etc. I also found the mixing of trends and correlation a little odd. I can see the rationale to relate those in the discussion. However, you

are showing trends, but then correlations with de-trended data. So, it feels a little inconsistent, though I understand you're trying to relate temperatures to the snow/ice changes. You might consider splitting the two into separate sections: first the trends, then the correlations and use the correlation to discuss the relationship between temperature and snow/ice on-off dates. Maybe that won't work well, but something to consider trying.

Thank you for this suggestion, we tried a few options for rearranging and have settled on separating the correlations into an introduction paragraph for each region, along with the mean phenology timing. Trends then follow with links back to the temperature/radiation data where suitable. Removing the numbers from the correlations and only mentioning the months improved the readability considerably.

Finally, more discussion of the data is needed (including proper citation as noted in the comments below), particularly data quality. IMS is described reasonably, but because IMS is an analyzed product and has used differing amounts and quality of data over the years, one should use caution when applying it towards trend analysis. I think it generally is okay in this context, looking at overall trends, but I think a comparison (at least for snow and sea ice) with passive microwave products would be beneficial to demonstrate consistency; at the very least, a discussion of the limitations of the dataset is needed.

Thank you for pointing out this omission on our part. A detailed comparison of IMS data for sea ice with microwave data was done by Brown et al 2014 – we showed that not only was IMS consistent with the overall patterns it was better than the traditional passive microwave as it can resolve the near coastal regions and leads when using the 4km product. Brown and Duguay, 2012, also highlight some limitations of IMS data when used for lake ice, and several other studies have commented on the utility of IMS for snow cover. These points have now been added to the data description.

Lines 128 – 141: IMS has been shown to outperform data from traditional passive microwave products (AMRS-E, SSM/I, SSMI-SSMIS) for both the timing and extent of first open water in the Arctic (Brown et al., 2014). For example, through the Barrow Strait in the CAA, the ability of the 4 km IMS data to resolve narrow channels lead to 17% more open water detected than with SSM/I, and 35% more open water than detected with AMSR-E, validated with RADARSAT-1 (Brown et al., 2014). Overall, most pixels compared between IMS and the two passive microwave datasets for first open water were within ± 5 days, with a greater percentage of the pixels in the categories beyond the ± 5 days identifying open water earlier with IMS than the other products (Brown et al., 2014). IMS has been shown to map higher snow cover fractions during the spring melt period than other snow products (Brown et al. 2010; Frei and Lee, 2010), but is reported to have mostly between 80-90% agreement with other snow products during the winter season of non-arctic North America, with better agreement in the later part of the winter season when deeper and more extensive snow cover is present (Chen et al., 2012). For lake ice, the 4 km IMS product occasionally identifies earlier lake ice-on dates in regions of prolonged cloud cover (e.g., northern Quebec, Canada), though both ice-on and -off timing detected using IMS are significantly correlated with, and comparable to, phenology dates extracted from the MODIS Snow Cover product (Brown and Duguay, 2012).

This goes doubly so for the temperature data. There needs to be more than a simple two-sentence paragraph. For one thing, reanalysis products are notorious for issues in polar regions because the data sparsity, particularly over sea ice. I note in comments below a couple artifacts that jumped out at me in the figures. And 2 m temperatures can be particularly problematic because the reanalysis models don't necessarily capture the surface boundary layer accurately. A now somewhat outdated paper (Lindsay et al., J. Climate, 2014) discusses some of the limitations and biases of different reanalysis products in the Arctic. There is nothing wrong with using ERA5 – it is one of the more updated and better products – but I think it is important to discussion the uncertainties and limitations. Along these lines, it might be interesting to correlation with 925 mb temperatures in addition to SAT (2 m) fields as that gets above the boundary layer and the reanalyses may be more reliable/consistent there.

ERA-Interim uses the observed 2 m temperatures in post processing procedures leading to smallest biases amongst other models (Dee et al., 2011, from Lindsay et al., 2014), presumably ERA5 does as well – though not confirmed in our literature search. ERA-Interim indicates a cold bias in Lindsay et al 2014 over land, however Wang et al 2019 show a warm bias over Arctic sea ice. Unfortunately, no recent 2 m temperature tests over arctic land areas could be located in our literature search - other useful and relevant information on ERA5 biases has been added instead:

Added lines 147-159: Temperature data near the surface (1000 mb) were reported to have a 0.89 K difference from radiosonde observations, and the ensemble spread is quite low at ~ 0.4 K or less, from 1979 – 2018, which can be used as an indicator of uncertainty (Hersbach et al., 2020). Compared to radiosonde temperature profiles in the Fram Strait, ERA5 had the smallest bias ($\leq$ 0.3°) and RMSE ($\leq$ 1.0°), and highest correlation coefficients ($\geq$ 0.96) over four other reanalysis datasets tested (ERA-Interim, JRA-55, MERRA-2, CFSv2) (Graham et al., 2019). The 2 m air temperature in ERA5 has improved fit to observations in the Arctic compared to its predecessor ERA-Interim (Hersbach et al., 2020), though Wang et al. (2019) show ERA5 has a warm bias over sea ice compared to observation data from buoys. The identified warm bias is stronger in the cold season, particularly when the 2 m air temperature is below -25°C (daily mean value of 5.4°C), however monthly mean differences between ERA5 and buoys are ~2°C or less through all months other than March, April, and May (Wang et al., 2019). Regionally, ERA5 performs best in the Central Arctic, followed by the Pacific Sector; the Atlantic sector shows good agreement only while the 2 m temperatures are above -25°C (Wang et al., 2019). We acknowledge that some small potentially spurious regions of opposite trend directions appear in some months of the temperature trend maps (e.g., February: Eastern Siberia, March and October: Arctic Ocean) however these data are not used in a quantitative comparison and therefore do not affect the overall discussion.

We redid the trend analysis and correlations using the 925 mb temperatures as suggested – although the overall trends maps are not overly different, the locations of some trend patterns did shift. Some potential artifacts are still evident at even that high above the surface, though not as strong and in different months than t2m. The correlations did not provide any extra insight over the 2m air temperatures (there were less significant correlations for sea ice and snow, but interestingly more for lake ice, which is something we may explore further in a separate ongoing lake ice project). Based on the readings regarding the handling of 2 m temperature in ERA-Interim, and the bias information discussed above, we feel 2 m temperature is more appropriate for representing the melt and freeze discussed here. A positive aspect of surface climate representation

ERA5 is the incorporation of improved lake parameterization – using the Flake model to couple with the atmosphere should be producing even better surface fluxes than previously – which have strong importance for melt/freeze. The 925 mb data was incorporated, however during the editing phase we opted to remove it as it did not add anything extra to the analysis other than length. Figure included below out of interest.

[Figure]

**Figure 8b –Trends in monthly 925 mb temperature from 2004 – 2019 in (a) January , (b) February, (c) March, (d) April € May, (f) June, (g) July, (h) August, (i) September, (j) October, (k) November, and (l) December.**

There are several fairly major comments here, along with the specific comments below. Thus, I recommend major revisions. However, I think the comments can be addressed with some thought and some rewriting and additional discussion, which I don't think should be too onerous. So, I think these revisions can be done in a reasonable amount of time.

**Specific Comments (by line number):**

56: I understand there are a lot of ice melt/freeze papers, and there is no need to be comprehensive, but it might be best to focus on more recent papers. E.g., Markus et al. (2009) has been updated by Stroeve et al., GRL, 2014, and even more recently by Bliss et al., Remote Sensing, 2017, and Bliss et al. Env. Res. Letters, 2019.

Noted, thank you, the missing recent papers have been added as suggested. Line: 56

102-109: I find the region definitions somewhat arbitrary and each encompasses quite different conditions. The Canadian Arctic spans the CAA, which has landfast and MYI, along with Baffin and Hudson Bays, which are nearly all FYI; and then it extends to a seemingly arbitrary longitude, east of the Canadian/Alaska border – so it not all of the Canadian Arctic, but most of it. The Alaska/Far East similar includes the Chukchi and some of the Beaufort, which have MYI and FYI, plus the Bering, which is exclusively FYI. Eurasia, seems like a catch-all region for everything else. And looking at ice/snow on-off dates is going to be at least partly latitude dependent, so splitting into regions with a wide range of latitude within them would seem to muddle any relationships. I agree this simplifies the analysis having only three regions, but I think explaining better the rationale would be helpful.

The snow and lake ice were grouped into the two continental regions to align with previous studies comparing Eurasia to North America, as they are affected by different weather patterns and storm tracks. The sea ice regions were reduced to three regions based on trend similarities and driving weather. We know the Canadian region is not changing the same way as the other Arctic regions in terms of trends. Eurasia is seeing earlier ice-off but the largest changes are in the US regions. Since there are 3 phenology parameters being compared the goal was to reduce the regions to encompass the broad patterns.  As an aside, investigating the trends based on FYI or MYI would certainly be interesting, but not possible with IMS data alone. Added the following for further explanation:

Lines 112-113: These regions were grouped based on similar trends in phenology parameters and differences in climate and weather characteristics at the hemispheric scale.

And similarly, is there a rational for using 56 N latitude as the southern boundary? It also seems somewhat arbitrary – it cuts off the southern Hudson Bay, for example. I guess it maybe encompasses the southern boundary of ice within the Bering, and it includes the large Arctic lakes. Thus, I can see a potential rationale (if I'm correct), but it should be explained within the text.

Correct, we were trying to maximize the ice included while minimizing the non-arctic land areas included.  Added the following sentence for clarity:

Lines 104-106: In this study, regions north of 56º were considered when examining pan-Arctic snow and ice phenology (Figure 1) to include much of the southern limits of the sea ice in the Bering Strait and large Arctic lakes that can be resolved using IMS.

111-114: Make sure you are clear on the source for the data. You mention NIC, but it has only recent data I believe. The historical archive is at NSIDC: https://nsidc.org/data/G02156/versions/1. This is referenced a few lines later for more information, but I think it is important to be clear upfront that the data were obtained from NSIDC (assuming they were). Also, the appropriate citation should be provided for the data set, as noted on the NSIDC page:

Most of the data was from NSIDC, but a few years were retrieved from NIC as we had trouble accessing the FTP at NSIDC. Text has been edited to put the emphasis on the NSDIC archive since the bulk of the data was obtained there. Dates included in each archive also added for clarity.

Lines 115-117: Snow and ice data were obtained from the Interactive Multisensor Snow and Ice Mapping System archived at the National Snow and Ice Data Centre (2004 – present, https://nsidc.org/data/G02156/versions/1) as well as from the National Ice Centre (2014 – present).

The NIC web address has recently changed, so it needs to be updated to: https://usicecenter.gov, or more specifically: https://usicecenter.gov/Products/ImsHome

Updated, thank you.

123: Similarly, for temperature data, more specificity is needed. Is there a website from where the data were obtained? And, if available, a full citation should be provided or at least the data set DOI (if one exists).

Thank you for pointing this omission out. The citations are in the reference list, but we inadvertently removed them from the data section of the manuscript. Added back in correctly now (CS3, 2017; Hersbach et al., 2020)

Lines 144: Temperature data are from the European Centre for Medium-Range Weather Forecasts (ECMWF) ERA5 global reanalysis (CS3, 2017 DOI: 10.24381/cds.f17050d7; Hersbach et al., 2020) and …

128: Maybe not directly relevant here, but could be worth mentioning that there is now a 1 km product starting in 2014. I wouldn't expect these to be used in the study as the time frame is too short to be of use, but readers may be interested to know that.

Clarified in the data section that the 1km product is not used here due to the limited time series available:

Line 119: … 1 km (2014 – present; not used in this study due to the limited time series),…

145: Were the data re-gridded/re-projected onto a consistent grid for comparison?

Data was analysed using centre-point locations (all points within the regions analysed), with ERA5 and APP-X reprojected into polar stereographic to match the IMS data.   Added the following to be explicit:

Lines 195-196: All data was projected to match the IMS data, and the centre point of all grid cells was used for the analysis.

198: Here is an example of where the paper could go farther. The authors note that freeze onset shows signification correlations with October air temperature. That's not surprising and it is not necessarily due to warmer air temperatures delaying freeze-up. As other studies have noted, it's actually the reverse: warm SSTs in the ice-free region give heat to the atmosphere and warm the atmosphere. So, it is the lack of sea ice that causes the October air temperatures.

This section of text has been removed in editing, but this is a good point that was not explicitly stated before – it has been worked into the freeze section with respect to the feedbacks also causing the downwelling longwave to increase during the fall

217-220: Not discussed here is the possible role of ice dynamics. How ice moves into and through the CAA can have a significant impact on the presence or lack of ice in the region during break-up and particularly freeze-up.

Line 278:  This is one of the main findings from Dauginis and Brown, 2020 that is cited here, but we failed to actually highlight the dynamics aspect. Added in parentheses: (due, at least in part, to increased ice dynamics through the CAA)

343-347: Another example where more analysis/discussion is needed. First snow-on dates are positively correlated with August, September, and October temperatures. But as noted above, the Aug, Sep, Oct temperatures may be the result of lack of sea ice causing the warming temperatures. And the lack of sea ice also provides a moisture source for earlier potential first snow. There is nothing groundbreaking in my analysis here, but it is not acknowledged at all in the manuscript.

Thank you for pointing this out, while we were inferring this in places where we mention increased atmospheric moisture you are quite correct that we never actually explicitly stated the open water / more moisture link. We have added several discussion sentences about the increased moisture in the atmosphere and resulting increase of snow, in particular for earlier snow-on in Euraisa.

Figure 1: Not to be too pedantic, but Greenland is shaded as Eurasia, but it is technically part of the North American plate. Just changing the color key to "Eurasia and Greenland" is reasonable, though it might look better shade Greenland with the North America color, or consider a separate shade altogether.

Greenland is now shaded the same as North America, thank you.

Figure 3: This is pretty obvious, but for total clarity, it would be worth adding a note in the caption that white indicates locations where there is no snow/ice on date (either because snow/ice never occurs there or snow/ice is present all year – e.g., GrIS and multi-year sea ice). Also, in the color legend, it would be helpful to label the center date (e.g., the pale yellow color). For snow/ice on date, it is okay because the middle is near day 365; but for snow/ice off labeling the yellow would be helpful for discerning colors between the extremes of 40 (red) and 275 (blue).

Centre data added to the colour legend, and addition line in the caption added: White regions indicate where either no snow/ice forms, or snow/ice remains all year, in 14 or more years of the dataset.

Figure 8: While listed in the captions, it would be more helpful to label each map with the month (or appropriate abbreviation, e.g., "Jan", "Feb", etc.), either in addition to or instead of the figure letter. This makes it easier to read the figure – I can just look at "Sep" instead of looking at "i)" and having to connect it to the month in the caption text.

Agreed, revised figure now included.

Figure 8j: This demonstrates some of my skepticism of SAT trends over sea ice. The "bullseye" of blue in the middle of the Arctic Ocean doesn't look realistic to me – probably an artifact of some interpolation (November shows a similar, albeit larger and more dispersed.

The October bull eye is an oddity, you can still see it in the 925mb layer though much less strong and the same direction as the surrounding ocean. However, this region does not have sea ice trends mapped as no consistent break up occurs here. We considered masking out the regions with no ice phenology in the temperature and longwave radiation maps, however decided against it as some interesting information would be lost, particularly through the Atlantic sector.

A sentence was added in the manuscript to acknowledge the potential artifacts in the data (both ERA5 and App-x) and the decision was made to continue with the 2m air temperature.

(ERA5) Lines 157-159: We acknowledge that some small spurious regions of opposite trend directions trends appear in some months the temperature trend data (e.g. February: Eastern Siberia; March: Arctic Ocean and October: Arctic Ocean) however these data are not used in a quantitative comparison so these small regions do not affect the overall discussion.

(APP-x) Lines 167-169: Some artificial patterns are evident in the data (e.g. March, Figure 9c, near the pole), however for the purpose of regional comparisons this is not limiting as this regions is not used in any quantitative comparisons.

Figure captions: Just a general comment on style – putting the figure letter - (a), (b), (c), etc. – at the end of the corresponding description is somewhat confusing and makes it more difficult to read. I think it would be easier to read if the letter is placed before the description for that letter.

Agreed. This has been changed in the captions now, thank you.

---

## Author Response (AR2)

Thank you to both reviewers, brief response below in blue text highlighting the changes made as requested

Reviewer2:
Excellent job responding to my comments. The manuscript is much improved in my view. The changes to Section 3.2 are particularly helpful. My only comment that I don't feel was fully addressed was on the IMS reference for the NSIDC product. As there is a recommended citation, with a DOI, I feel that that should be used, not just the website address for the product. The recommended citation is:

U.S. National Ice Center. 2008, updated daily. IMS Daily Northern Hemisphere Snow and Ice Analysis at 1 km, 4 km, and 24 km Resolutions, Version 1. [Indicate subset used]. Boulder, Colorado USA. NSIDC: National Snow and Ice Data Center. doi: https://doi.org/10.7265/N52R3PMC. [Date Accessed].

I'll leave it to the editor to decide how strictly to enforce this, but in my view, citation with DOIs should always been done when available. This greatly improves proper recognition for the data producers and increases discoverability by readers.

Thank you, this citation has already been used in the references, however we have now modified the text to reflect that directly.

Reviewer 3
Review of tc-2021-52
Recent changes in Pan-Arctic sea ice, lake ice, and snow on/off timing
Dauginis and Brown

General comments
The study presents a thorough look at sea ice, lake ice and snow cover trends over the Arctic, as seen by IMS 24 and 4 km products. Regional as well and Pan-Arctic trends are discussed. While individual trends of each of these parameters have been examined in previous works, the novelty of the paper lies in assessing these together with a common methodology and format. Interconnections as well as the relevance of air temperature and downwelling longwave radiation to the detected trends are analyzed. Referencing seems to be thorough and accurate, although I note some cases where more recent work could be used.
In my view the authors have provided a satisfactory response to concerns brought up after the original submission. Although I did not review the original work, I have no other major issues to note. My only major gripe is with Section 3 which is very long but perhaps this cannot be helped. Some minor suggestions and typos are listed below.

Minor/editorial comments
1. line 33: "Derksen et al., 2012" (typo)
Now correctly listed, Derksen et al. 2012

2. line 66: some more recent work on SWE retrieval from PWM could be cited, e.g. Mudryk et al., 2015; Larue et al., 2017; Pulliainen et al., 2020

Thank you, kept the older Derksen et al reference and added in Mudryk et al and Pullianinen et al

3. lines 82-84: The moderate temporal coverage of SAR systems is an old criticism and disregards recent advancements (e.g. Sentinel-1, RCM, and many commercial SAR providers). e.g. your citation of Howell et al. (2019) used only RadarSat-1 and -2. New systems naturally cannot help with historical trends, but it should somehow be acknowledged that the situation has much improved especially regarding contemporary sea ice monitoring in the Arctic.

Agreed. Reference to improved SAR coverage through RCM in particular was at one point in the manuscript, but appears to have been removed through our various edits – sentence now revised to:

SAR estimates of snow and ice cover provide the highest spatial resolution compared to other products, and while previously limited by the moderate temporal resolution, narrow swath width, and limited image availability across the Arctic (Brown et al. 2014; Howell et al. 2019), recent advances through additional sensors (e.g. Radarsat Constellation Mission, Sentinal-1) have much improved both temporal and spatial coverage as well as data availability.

4. line 88: you cite IMS as an "alternative approach to snow and ice mapping". However, I'd argue IMS combines many of the previously described approaches, adding human interference and interpretation in the mix. Some other wording could be used in order not to confuse a potential future non-expert reader?

Revised to: "A combined approach to snow and ice mapping is possible with the use of …"

5. line 192: "All data were"
Corrected, thanks

6. line 452; "correlations" (typo)
Corrected, thanks

7. line 452 (and 488, maybe elsewhere): "downwelling longwave radiation" (radiation was missing). Maybe introduce an acronym? The term is used quite frequently.

LW↓ is selected for 'downwelling longwave radiation at the surface' and now used throughout (with the exception of figures and tables where the full wording was left for completeness).

8. Figures 4 and 5: All panels with temporal trends except Fig 4c and f show the timeline between 1997 and 2019. I would suggest adjusting the x-axis in Fig 4c&f to match the other

panels, despite the lack of data on FOW/WCI and FO/CIC prior to 2004. It would make comparing the figures easier and highlight that the temporal span in Fig4c/f is indeed shorter.

Very good point, figure axes have now been modified to match the rest.